# A small molecule that induces translational readthrough of *CFTR* nonsense mutations by eRF1 depletion

Jyoti Sharma[1,2,3,12], Ming Du[1,4,12], Eric Wong[5], Venkateshwar Mutyam[1,2], Yao Li[1,2], Jianguo Chen[1,2], Jamie Wangen [6], Kari Thrasher[1,4], Lianwu Fu[1,7], Ning Peng[1,2], Liping Tang[1,2], Kaimao Liu[1,4], Bini Mathew[8], Robert J. Bostwick[8], Corinne E. Augelli-Szafran[8], Hermann Bihler [5], Feng Liang[5], Jerome Mahiou[5], Josef Saltz[5], Andras Rab[9], Jeong Hong[9], Eric J. Sorscher[9], Eric M. Mendenhall [10], Candice J. Coppola[10], Kim M. Keeling [1,4], Rachel Green [6], Martin Mense [5], Mark J. Suto[8], Steven M. Rowe [1,2,11,13] & David M. Bedwell [1,4,13✉]

Premature termination codons (PTCs) prevent translation of a full-length protein and trigger nonsense-mediated mRNA decay (NMD). Nonsense suppression (also termed readthrough) therapy restores protein function by selectively suppressing translation termination at PTCs. Poor efficacy of current readthrough agents prompted us to search for better compounds. An NMD-sensitive NanoLuc readthrough reporter was used to screen 771,345 compounds. Among the 180 compounds identified with readthrough activity, SRI-37240 and its more potent derivative SRI-41315, induce a prolonged pause at stop codons and suppress PTCs associated with cystic fibrosis in immortalized and primary human bronchial epithelial cells, restoring CFTR expression and function. SRI-41315 suppresses PTCs by reducing the abundance of the termination factor eRF1. SRI-41315 also potentiates aminoglycoside-mediated readthrough, leading to synergistic increases in CFTR activity. Combining readthrough agents that target distinct components of the translation machinery is a promising treatment strategy for diseases caused by PTCs.

---

[1] Gregory Fleming James Cystic Fibrosis Research Center, University of Alabama at Birmingham (UAB), Birmingham, AL, USA. [2] Department of Medicine, University of Alabama at Birmingham (UAB), Birmingham, AL, USA. [3] Department of Microbiology, University of Alabama at Birmingham (UAB), Birmingham, AL, USA. [4] Department of Biochemistry and Molecular Genetics, University of Alabama at Birmingham (UAB), Birmingham, AL, USA. [5] CFFT Lab, Cystic Fibrosis Foundation, Lexington, MA, USA. [6] Department of Molecular Biology and Genetics and Howard Hughes Medical Institute, Johns Hopkins University School of Medicine, Baltimore, MD, USA. [7] Department of Cell, Developmental and Integrative Biology, University of Alabama at Birmingham (UAB), Birmingham, AL, USA. [8] Southern Research, Birmingham, AL, USA. [9] Department of Pediatrics, Emory University, Atlanta, Georgia. [10] Department of Biological Sciences, The University of Alabama in Huntsville, Huntsville, AL, USA. [11] Department of Pediatrics, University of Alabama at Birmingham (UAB), Birmingham, AL, USA. [12]These authors contributed equally: Jyoti Sharma, Ming Du. [13]These authors jointly supervised this work: Steven M. Rowe, David M. Bedwell. ✉email: dbedwell@uab.edu

Cystic fibrosis (CF) is an autosomal recessive disease affecting over 70,000 people worldwide[1]. Over 2000 variants of the cystic fibrosis transmembrane conductance regulator (CFTR) gene are known[2–4]; 350 of which are confirmed to be disease-causing by impeding CFTR cell surface expression and/or compromising its anion channel function[5]. Clinical manifestations include mucus accumulation, chronic respiratory infection, pancreatic insufficiency, and male infertility, leading to severe morbidity and early mortality[6]. CFTR modulator therapy using correctors and potentiators has proven transformative for many CF patients who retain at least partial expression of full-length CFTR protein[7–9]. However, ~11% of CF patients carry a nonsense mutation, which generates a premature termination codon (PTC) in the CFTR mRNA, leading to the generation of a truncated CFTR protein that cannot be affected by current modulator therapies. Moreover, PTC-containing CFTR mRNAs are often degraded by nonsense-mediated mRNA decay (NMD), severely reducing the pool of CFTR mRNA available for translation[10,11]. Together, these PTC-mediated events often lead to total abrogation of CFTR function and are associated with severe CF manifestations[12]. Many other genetic diseases, including Duchenne muscular dystrophy, β-thalassemia, and numerous types of cancers, are also caused by nonsense mutations[13].

Suppressing translation termination selectively at PTCs (also referred to as readthrough) is a potential treatment strategy for genetic diseases caused by nonsense mutations, including CF[13]. Readthrough agents suppress termination at in-frame PTCs by promoting the accommodation of near-cognate aminoacyl-tRNAs into the ribosomal acceptor (A) site. The insertion of an amino acid into the nascent polypeptide at the site of the PTC allows translation elongation to continue in the correct reading frame and generate a full-length protein[14–16]. The prototypical nonsense suppression approach was first demonstrated using aminoglycosides, which bind to a region of the 18S eukaryotic rRNA known as the decoding center and reduce proofreading of the A-site by the ribosome[17]. Proof-of-principle studies show that aminoglycosides can induce readthrough in a variety of disease contexts, including CF, where a subset of aminoglycosides restore partial CFTR activity in vitro and in vivo[18–24]. However, conventional aminoglycosides, which are normally indicated clinically for their antimicrobial effects, exhibit modest readthrough efficacy and are unsuitable for long-term use due to off-target effects that lead to toxicity[25,26]. Thus, there is an urgent need for safe, effective PTC suppression agents that achieve sufficient efficacy to confer clinical benefit to patients with CF and other diseases caused by nonsense mutations[15,27–29].

Translation termination is a multi-step process involving multiple proteins, suggesting that distinct therapeutic targets may be available to mediate more effective PTC suppression[30,31]. For example, readthrough efficiency may be enhanced by suppressing the recruitment of termination factors[32,33] to increase the probability of near-cognate tRNA accommodation at PTCs. In the current study, we examine a chemical series discovered through a high throughput screen, SRI-37240 and its derivative SRI-41315, for their ability to suppress CFTR nonsense mutations. These agents promote readthrough by prolonging the translational pause that occurs at PTCs by reducing the abundance of the termination factor, eRF1, a novel mechanism for pharmacologically induced readthrough that is synergistic with aminoglycosides and translates to predictive preclinical models.

## Results

### Identification of compounds that suppress nonsense mutations using novel NanoLuc-based reporters. Current nonsense

suppression agents are unable to restore sufficient protein function to be an effective treatment for many genetic diseases, including CF. A disadvantage of many reporters previously used to identify readthrough compounds is that they do not undergo NMD like most endogenous PTC-containing mRNAs, and therefore, are unlikely to accurately reflect the level of protein activity restored via readthrough of endogenous transcripts. To address this weakness, we built an NMD-sensitive readthrough reporter that is more likely to reflect the protein activity generated by readthrough of PTCs in mRNAs subject to NMD. This dual RT/NMD reporter (Fig. 1a) is composed of a NanoLuc cDNA containing a PTC (UGAA) at codon W134 that is fused to human beta-globin containing two intronic sequences that have been shown to induce classical, exon junction complex (EJC)-mediated NMD[34]. Characterization of this reporter indicated that it responds to both readthrough and NMD inhibition (Suppl. Fig. 1a-c). This reporter was stably expressed in Fischer rat thyroid (FRT) cells. Based on the basal signal output and the response to readthrough induced by the aminoglycoside G418 (Suppl. Fig. 1a), a monoclonal FRT reporter line was selected to carry out a high throughput screen (HTS).

771,345 small molecules were screened using the RT/NMD FRT reporter cell line. Thirty-two negative control (cells only) and thirty-two positive control (cells treated with 300 µg/ml G418) wells were included on each plate in the HTS. In addition to normalizing test compound data, the control data was also used to monitor assay performance by determining the Signal to Background Ratio (S/B), Signal to Noise Ratio (S/N), the percent Coefficient of Variance (% CV) of controls, and the Z′-factor (Suppl. Table 1). The Z′-factor is an index of assay performance, and values >0.5 is an indication of assay reliability. In these assays obtained throughout the screen, the Z′-factor was 0.78. The Statistical Threshold of Activity, defined as the % Activation mean of all test data plus three times the % Activation standard deviation of negative controls, was 13.57% (Fig. 1b).

After the initial screen, a total of 2004 compounds were selected to confirm activity by retesting them for a concentration-dependent response in the dual RT/NMD reporter assay used for the HTS (Suppl. Fig. 2). Of these, 180 were confirmed as hits with Activation Emax >20% of the G418 controls. Fresh samples were obtained for 148 of these compounds for further testing. SRI-37240 was among the most active compounds with an Emax = 718% of the G418 control (Fig. 1c). Importantly, SRI-37240 did not show any effect on a wild-type NanoLuc (PTC-independent) reporter (Suppl. Fig. 1d), demonstrating that SRI-37240 acts in a PTC-specific manner. We also examined the effect of combining SRI-37240 with G418 in FRT cells expressing the NanoLuc dual RT/NMD reporter. Co-treatment of SRI-37240 with G418 (100 µg/ml) in FRT reporter cells resulted in an enhancement of NanoLuc activity, suggesting that SRI-37240 and G418 likely induce readthrough by independent mechanisms (Fig. 1d).

### SRI-37240 restores CFTR expression and function in vitro and synergizes with G418. We next determined whether SRI-37240 could suppress PTCs associated with cystic fibrosis. To test whether SRI-37240 induced sufficient readthrough to restore therapeutically meaningful CFTR protein and function, we performed CFTR-specific electrophysiological and biochemical assays in FRT cells stably expressing a human CFTR cDNA containing the G542X (UGAG) nonsense mutation or the wild-type control cDNA. SRI-37240 induced concentration-dependent increases in CFTR-dependent (forskolin-stimulated and sensitive to the inhibitor $CFTR_{Inh}$-172) chloride conductance ($0.03 \pm 0.00$ to $0.60 \pm 0.01$ mS/cm$^2$, $p < 0.0001$) (Fig. 2a). In addition, both 10 and 30 µM SRI-37240 exhibited strong synergy with 100 µg/ml G418 ($0.97 \pm 0.02$ and $2.8$

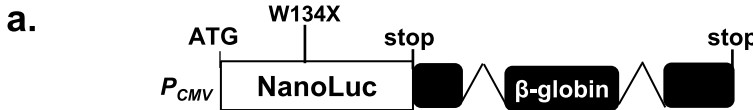

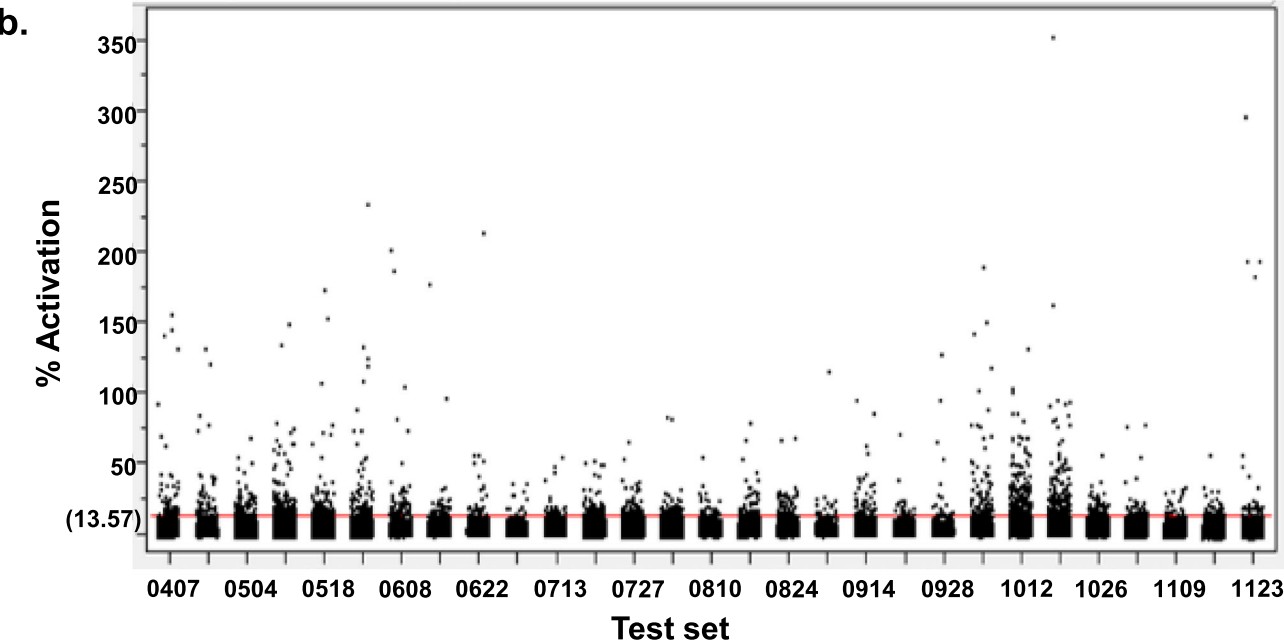

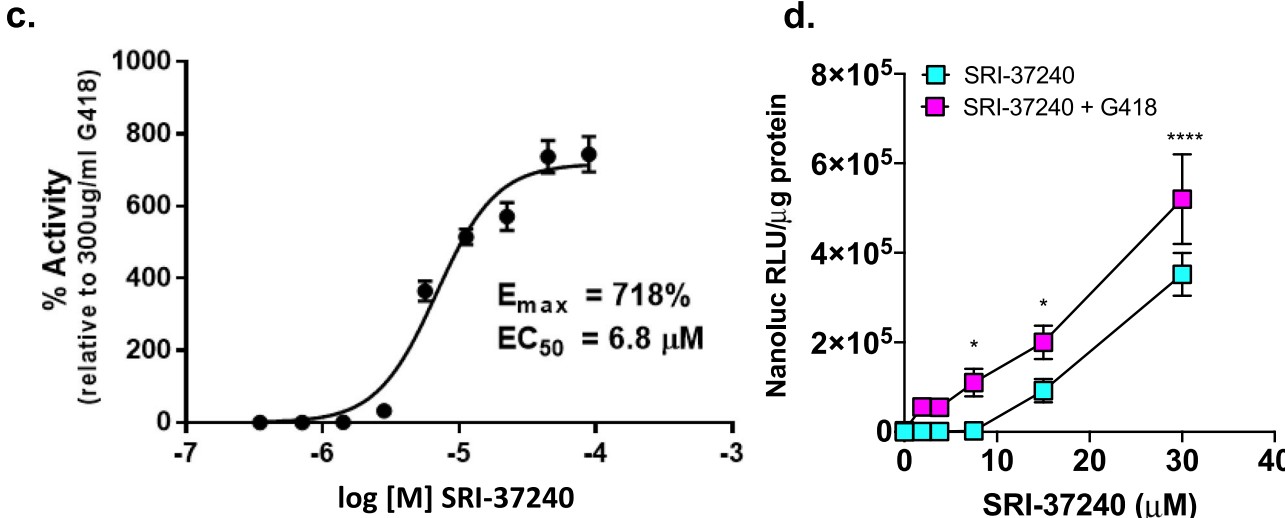

**Fig. 1 Identification and characterization of novel readthrough agents using an RT/NMD NanoLuc reporter in FRT cells. a** The reporter used to perform HTS. **b** Scatter plot of HTS results using the reporter cell line. The screen was comprised of 29 test sets. Hit compounds were defined as those with activity ≥ mean ± 3 SD of all test compound data (13.57%). Six data points are off-scale (460–2900%) and are not shown. Concentration-response data for **c** SRI-37240 alone (mean of 2 replicates, bars indicate range) and **d** comparison of SRI-37240 alone and combined with 100 μg/mL G418 in FRT cells expressing the NanoLuc RT/NMD reporter. In panel **c**, the NanoLuc activity is expressed as the percent (mean ± SD) of the signal generated by the positive control (300 μg/mL G418). In panel **d**, the NanoLuc activity is normalized to total cellular protein. Two-way ANOVA was used to compare NanoLuc activity (mean +/- SD) in cells treated with both SRI-37240 and G418 to cells treated with SRI-37240 alone ($n = 4$; *$p < 0.05$ and ****$p < 0.0001$). Source data is available as a source data file.

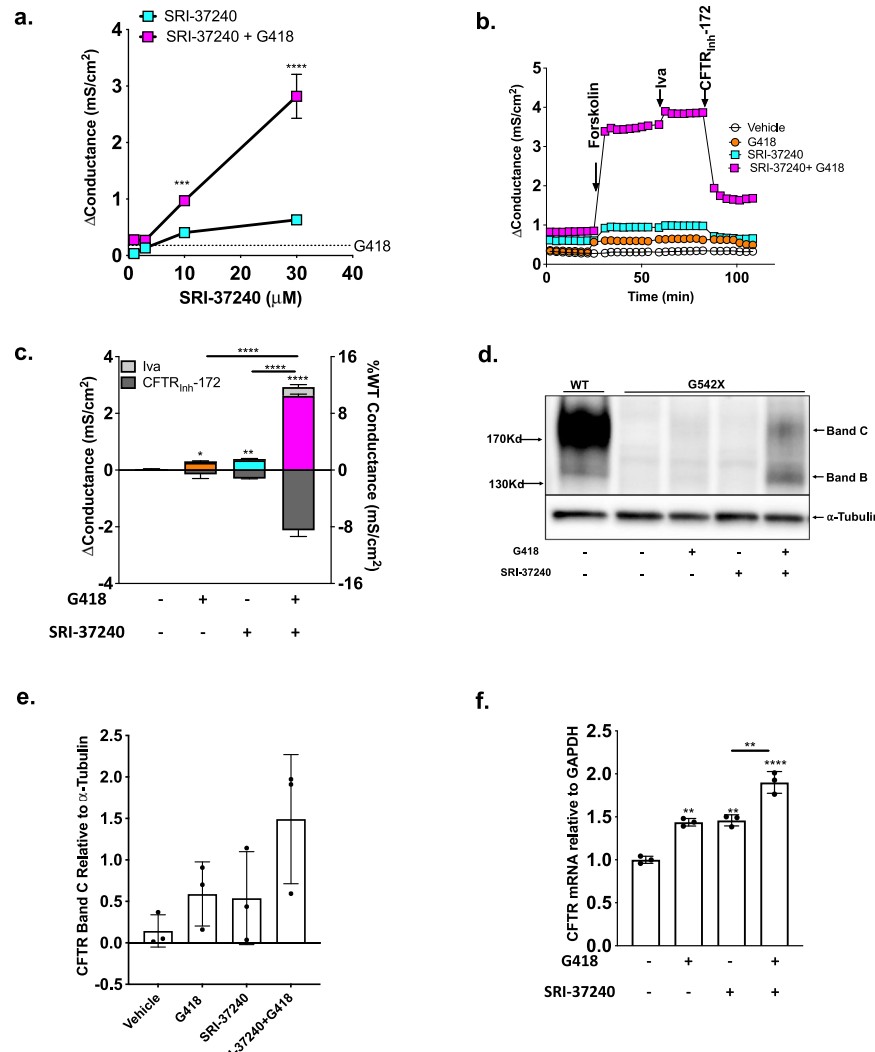

**Fig. 2 CFTR expression and function is enhanced after combination treatment with SRI-37240 and aminoglycosides in FRT cells stably transduced with human *CFTR*-G542X cDNA.** FRT monolayers were treated with SRI-37240 (10 μM unless indicated otherwise)+/−G418 (100 μg/mL) for 48 h on a liquid–liquid interface. **a** Forskolin-induced CFTR conductance after treatment with SRI-37240 at 1, 3, 10, and 30 μM concentrations alone (blue squares) and in combination with G418 (purple squares) ($n = 4$, data are statistically analyzed using Two-way ANOVA followed by Sidak's post hoc test, $p = 0.0006$ (SRI-37240 vs SRI-37240 (10 uM) + G418), $p < 0.0001$ (SRI-37240 vs SRI-37240 (30 uM) + G418). **b** Representative conductance tracings after treatment with SRI-37240+/−G418. **c** Corresponding summary of conductance measurements of SRI-37240+/−G418 after forskolin stimulation (represented by corresponding colors as in raw tracings in Fig. 2b) and ivacaftor (iva) (light gray) induced CFTR conductance and CFTR$_{Inh}$-172 (dark gray) mediated inhibition; also expressed as % of wild-type CFTR conductance in isogenic FRT cells (i.e., 25 mS/cm$^2$) ($n = 3$, data are expressed as mean ± S.D. and statistically analyzed using repeated measures one-way ANOVA followed by Tukey's post hoc test). ****$p < 0.0001$ (G418 vs SRI-37240 + G418), ****$p < 0.0001$ (SRI-37240 vs SRI-37240+/−G418). **d** Representative western blot showing increased CFTR band C (fully mature) and band B (core glycosylated) after combination treatment with SRI-37240+/−G418. Total protein loaded: WT = 10 μg and G542X = 30 μg). **e** Densitometric quantification of full-length CFTR Band C (mature form) relative to α-tubulin from FRT cells expressing wild-type or G542X *CFTR* cDNAs after treatments with SRI-37240 and G418 (alone or in combination). **f** Steady-state *CFTR* transcript levels by real-time RT-PCR relative to *GAPDH* after treatment with SRI-37240+/−G418. Data is the representation of $n = 3$ monolayers per condition and 2–4 experimental repeats for each experiment ($n = 3$, data are expressed as mean ± S.D. and statistically analyzed using ordinary one-way ANOVA followed by Tukey's post hoc test, $p = 0.0019$ (Vehicle vs G418), $p = 0.0014$ (Vehicle vs SRI-37240), $p < 0.0001$ (Vehicle vs SRI-37240 + G418), $p = 0.002$ (SRI-37240 vs SRI-37240 + G418). For all panels, *$p \leq 0.05$, **$p \leq 0.01$, ***$p \leq 0.001$, ****$p \leq 0.0001$. Source data is available as a source data file.

± 0.22 mS/cm$^2$, respectively, $p < 0.0001$) (Fig. 2a, electrophysiologic tracings in Suppl. Fig. 3a). When combined with G418 (100 μg/ml), SRI-37240 (10 μM) increased forskolin-stimulated conductance (G$_t$) to 10.5% of wild-type compared to SRI-37240 (1.4%) or G418 (1%) alone (Fig. 2b, c).

Consistent with the CFTR functional data, western blotting showed that SRI-37240 combined with G418 significantly increased the amount of Band C CFTR protein, which represents the full-length, fully glycosylated form of CFTR and Band B, which represents the unprocessed, immature form of full-length CFTR protein (Fig. 2d, e). Quantitation of band C CFTR protein indicated that when SRI-37240 was combined with G418, ~25% of the wild-type level of full-length CFTR protein was restored compared to cells treated with SRI-37240 (~6%) or G418 (~9%) alone (Fig. 2d, e).

While we observed a modest (<twofold) increase in forskolin-stimulated CFTR conductance in FRT cells expressing wild-type

CFTR following treatment with SRI-37240, the magnitude of improvement was small compared to CFTR-G542X FRT cells and was not synergistic with G418 (Suppl. Fig. 3b-c), thereby eliminating non-specific effects as the cause for the increases in CFTR expression and function with SRI-37240+/−G418. As an additional test of specificity, we did not observe a meaningful increase in cAMP-dependent conductance following the addition of SRI-37240+/−G418 in parental FRT cells that did not carry a CFTR cDNA (Suppl. Fig. 3d).

Next, we examined the effect of SRI-37240 on CFTR mRNA abundance in G542X FRT cells, which is not subject to EJC-mediated NMD due to its expression from a cDNA construct lacking introns[15]. Both SRI-37240 and G418 alone induced modest increases in CFTR mRNA levels (1.4-fold of vehicle each), that slightly increased when used in combination (1.8-fold; $p < 0.01$, $p < 0.0001$, respectively) (Fig. 2f). Together, these results indicate that for FRT cells expressing a CFTR cDNA construct, SRI-37240 restored CFTR activity primarily through readthrough, although a modest stabilization of the CFTR mRNA also occurred.

In combination with G418, SRI-37240 also demonstrated relatively broad activity for several other CFTR nonsense mutations beyond G542X (R553X, R1162X, and W1282X) representing various sequence contexts (Supplementary Notes; Suppl. Fig. 4). The action of SRI-37240 was most pronounced with the W1282X context, where residual CFTR activity is generated by the truncated peptide, augmenting its apparent rescue with both nonsense suppression and CFTR modulators[15]. In all cases, the effect of readthrough combinations was further augmented by addition of the CFTR corrector lumacaftor (Suppl. Fig. 4).

**SRI-37240 alters cellular translation termination at PTCs as determined by ribosomal profiling**. To examine the effect of SRI-37240 on the efficiency of global translation termination, we performed ribosome profiling[35]. HEK293T cells were treated with SRI-37240 for 24 h. DMSO or G418 treated cells served as negative and positive readthrough controls, respectively. We then examined whether global readthrough at normal termination codons (NTCs) occurred in these cells by performing a metagene analysis, aligning all transcripts by their stop codons (Fig. 3a). In agreement with previous data[36], G418 (orange) stimulated genome-wide readthrough of natural stop codons, resulting in decreased density of ribosomes at termination codons and a corresponding increased density of ribosomes in 3′-UTRs relative to DMSO-treated cells (black). In contrast, SRI-37240 (blue) increased global densities of ribosomes at normal stop codons without affecting densities of ribosomes in 3′-UTRs (Fig. 3a). We next quantified the readthrough effect of SRI-37240 at the level of individual transcripts. First, we calculated a stop codon pause score for every transcript by determining the ratio of the ribosome density at the termination codon relative to the ribosome density in the coding region (Fig. 3b). Consistent with the metagene gene analysis, G418 treatment decreased stop codon pause scores while SRI-37240 treatment increased stop codon pause scores relative to DMSO-treated cells ($p = 1.4 \times 10^{-248}$ and $p = 8.6 \times 10^{-302}$, respectively, Wilcoxon signed-rank test), demonstrating that these effects on translation termination occur globally and are not driven by a subset of transcripts. Second, we calculated ribosome readthrough scores (RRTS) for each transcript by dividing the total ribosome density in the region of the 3′-UTR between the normal termination codon and first in-frame termination codon by the total ribosome density of the coding sequence (Fig. 3c). As expected, G418 treatment globally increased readthrough by this metric while SRI-37240 did not

increase readthrough compared to the DMSO control ($p = {\sim}0$ and $p = 0.11$, respectively, Mann–Whitney U). Taken together, these data suggest that SRI-37240 mediates readthrough through a mechanism that is distinct from G418. Unlike G418, SRI-37240 induces a prolonged pause at stop codons and inhibits translation termination at PTCs without stimulating readthrough at NTCs above basal levels.

**Establishing a structure-activity relationship (SAR) with SRI-37240 derivatives**. To explore whether analogs of SRI-37240 could be generated that improved readthrough potency and efficacy beyond that of the parental scaffold, we synthesized over 40 compounds based upon the structure of SRI-37240 (Fig. 4a) and determined their ability to induce readthrough. SRI-41315 (Fig. 4b) is an analog of SRI-37240 that exhibited better physio-chemical features (Table 1). Both compounds exhibited target cell cytotoxicity (CC50) values >50 μM in both FRT and 16BE14o- cells (Table 1). We next compared the readthrough activity of SRI-41315 to its parent compound, SRI-37240. Because read-through efficiency and the rescue of CFTR function can vary among different cell lines, and human cell lines may be more predictive of clinical utility[37,38], we expressed the dual RT/NMD NanoLuc reporter not only in FRT cells, but also in human-derived 16HBE14o- parental cells[6,39] to examine readthrough efficiency. In FRT cells, SRI-41315 showed a peak fold-increase in NanoLuc activity (2343 ± 152 relative to vehicle at 30 μM) that was substantially greater than that produced by SRI-37240 (354 ± 24 relative to vehicle at 30 μM; Fig. 4c), indicating the markedly improved efficacy of SRI-41315. SRI-37240 exhibited minimal readthrough activity in 16HBEo- cells expressing the RT/NMD NanoLuc reporter, while SRI-41315 demonstrated much larger, dose-dependent fold-increases in NanoLuc activity, peaking at 186 ± 25 relative to vehicle (Fig. 4d). When SRI-37240 was combined with G418 (50 μg/ml, a sub-optimal concentration in 16HBE14o- cells that did not induce toxicity), strong dose-dependent synergy was observed, peaking at 200-fold over vehicle at 30 μM, well above that observed with G418 alone (50-fold over vehicle; Fig. 4e). The level of readthrough achieved by SRI-41315 was also synergistically enhanced by the addition of G418, with a 350-fold increase in 16HBE14o- reporter response observed relative to vehicle. Together, these results indicated that the improved potency and efficacy of SRI-41315 was also observed in 16HBE14o- cells in a manner that could be further enhanced when combined with G418.

**SRI-41315 rescues CFTR function via readthrough and is enhanced by G418**. To determine the extent of CFTR activity restored by SRI-41315 alone or in combination with G418, we initially used FRT cells stably expressing a CFTR G542X cDNA to assay CFTR-mediated conductance. In CFTR-G542X FRT cells, SRI-41315 increased forskolin and ivacaftor-stimulated conductance ($G_t$) (Suppl. Fig. 5a-b), and addition of G418 resulted in significantly higher efficacy than either treatment alone, which is further confirmed by western blot analysis (Suppl. Fig. 5c-d).

We next used a gene-edited CFTR-G542X (16HBEge) cell line[40] to assay CFTR-mediated short-circuit currents. The 16HBEge cell line serves as a more physiologically relevant in vitro CF model because it carries a single copy of the genomic CFTR gene that harbors the G542X nonsense mutation, and also contains its natural intronic structure and is expressed via its native CFTR promoter. In contrast to the heterologous CFTR cDNA-based expression system in FRTs that is not subject to EJC-dependent NMD, the CFTR-G542X mRNA in the 16HBEge cell line is subject to NMD[40]. In CFTR-G542X 16HBEge cells, SRI-41315 alone induced a modest increase in forskolin plus

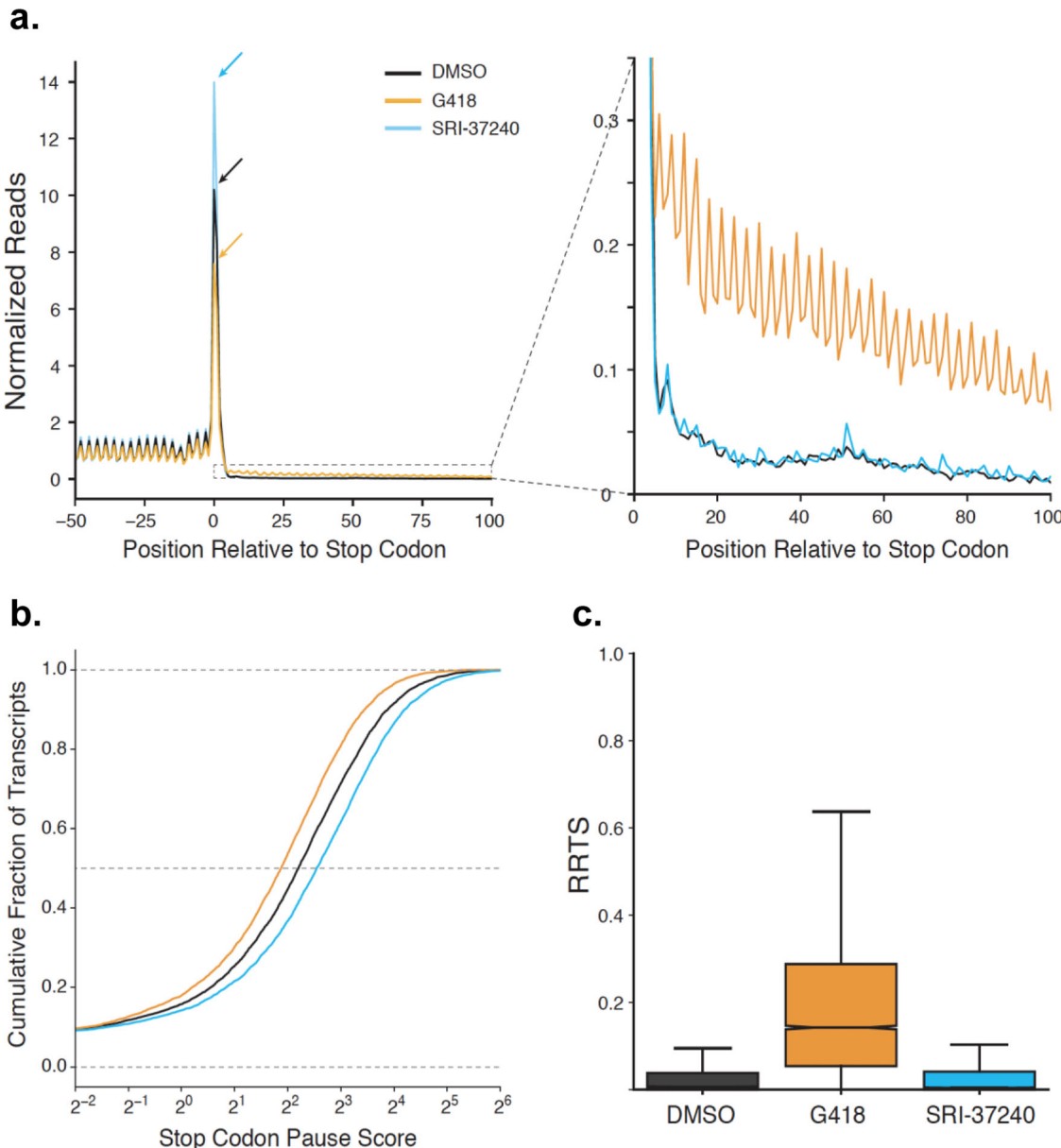

**Fig. 3 Exploring the effect of SRI-37240 on global cellular translation termination using ribosome profiling.** HEK293T cells were treated with 0.1% DMSO (black), 0.5 mg/mL G418 (orange), or 10 μM SRI-37240 (cyan) for 24 h. **a** Metagene plots showing normalized ribosome densities relative to the position of the termination codon at position 0. Arrows indicate the peak for ribosomes at the stop codon to facilitate comparison (left). A magnified view of 3′UTRs allows comparison of translation in this region (right). **b** Stop codon pause scores were calculated for each transcript and compared using cumulative histograms. G418 globally decreased ribosome density at stop codons ($P = 1.4 \times 10^{-248}$) while SRI-37240 increased ribosome density at stop codons ($P = 8.6 \times 10^{-302}$, two-sided Wilcoxon signed-rank test). **c** RRTS values for all transcripts are compared between treatments using box and whisker plots. Median values are indicated with the notch showing 95% confidence intervals and whiskers indicate 1.5 times the interquartile range. Outliers are excluded from the plot. G418 increased RRTS values while SRI-37240 did not significantly increase RRTS values while SRI-37240 did not significantly increase RRTS values relative to DMSO ($P < 0.0001$ and $P = 0.11$ respectively, one-sided Mann–Whitney $U$). Source data is available as a source data file.

ivacaftor-stimulated current (AUC 14.7 ± 7.7 μA/cm²*min vs. 0.0 ± 0.8 μA/cm²*min, control, $p = 0.0541$) (Fig. 5a, b). In contrast, SRI-37240 was inactive when used alone, which is consistent with the lack of readthrough observed in 16HBE14o- cells expressing the RT/NMD NanoLuc reporter. When SRI-41315 was combined with G418 in G542X 16HBEge cells, CFTR function increased substantially (AUC 116.0 ± 7.7 μA/cm²*min vs. G418 alone 55.0 ± 10.0 μA/cm²*min; $p < 0.0001$) (Fig. 5a, b) while maintaining transepithelial electrical resistance (TEER) (Suppl. Fig. 5e). This level of CFTR activity was ~7% of wild-type levels (wild-type 16HBE cells exhibited an AUC of 1878 ± 55.9 μA/cm²*min; for

data tracings, see Suppl. Fig. 6a-b). As a clinically meaningful comparator of CFTR modulator therapy, this level of CFTR function is ~19% of that generated in F508del 16HBEs treated with lumacaftor plus ivacaftor (see Suppl. Fig. 6c-d). In contrast, *CFTR*-G542X 16HBEge cells treated with both SRI-37240 and G418 did not show more CFTR function than cells treated with G418 alone (AUC 61.0 ± 10.1 μA/cm²*min, $p =$ ns Fig. 5a, b). Immunoblotting of *CFTR*-G542X 16HBEge cells confirmed the restoration of full-length, mature CFTR (band C) with SRI-41315 plus G418 combination therapy (~8% of wild-type CFTR) (Fig. 5c, d), whereas treatment with either SRI-41315 or G418 alone

**a.  SRI-37240**

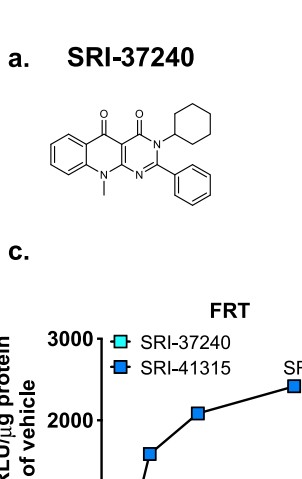

**b.  SRI-41315**

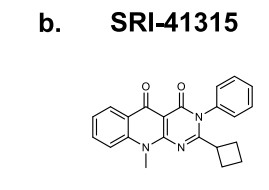

**c.**

**d.**

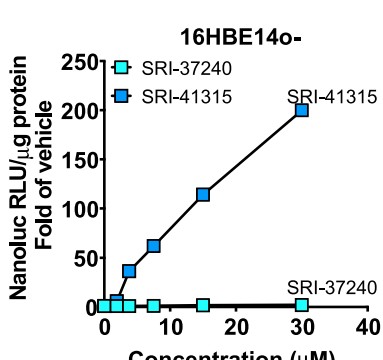

**e.**

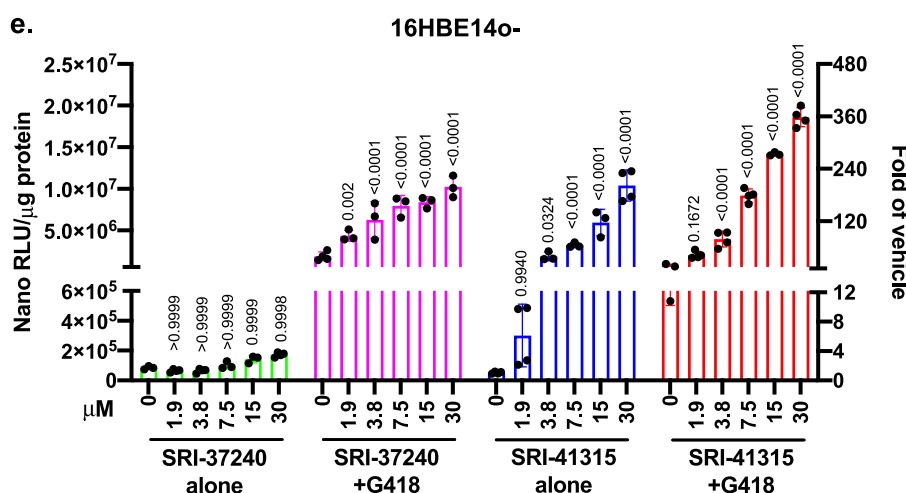

**Fig. 4 SRI-41315, a structural analog of SRI-37240, showed improved potency and efficacy in FRT cells that translated to 16HBE14o- cells. a** SRI-37240 structure. **b** SRI-41315 structure. The average fold-increase in relative luminescence unit (RLUs) in **c** FRT cells or **d** 16HBE14o- cells stably expressing the W134X NanoLuc RT/NMD reporter after 24-h (FRTs) or 48-h treatment (16HBEs) of treatment with increasing concentrations (in μM) of SRI-37240 or SRI-41315. Average RLUs from 3 or 4 independent replicates were normalized to the average RLUs from cells treated with vehicle alone (0.3% DMSO). **e** NanoLuc RLUs (normalized to total lysate protein) from 16HBE14o- cells expressing the W134X NanoLuc RT/NMD reporter that were treated with SRI-37240 and SRI-41315 alone (at various concentrations shown in μM) alone or in combination with G418 (50 μg/mL). Each column indicates the mean +/−SD ($n = 3$ or 4). Exact $p$ values (shown above the columns) were determined by a two-way ANOVA by comparing each treatment group to the untreated control with each cohort. All of the assays were independently repeated with similar results. Source data is available as a source data file.

restored less CFTR protein (~4% of wild-type CFTR). Taken together, these results indicate that SRI-41315 restored CFTR function more efficiently in *CFTR*-G542X HBEge cells than SRI-37240, and its activity was further enhanced by G418.

**SRI-41315 induces translational readthrough by depleting eRF1 protein levels.** Initial experiments using a previously established in vitro transcription/translation system[41] showed that G418, which acts by reducing ribosomal proofreading during translation, showed a dose-dependent increase in readthrough while SRI-41315 did not (Suppl. Fig. 7a). This suggested that this compound may not act directly during the translation process. To better understand how

SRI-41315 exerts its effect on translation termination, we examined the abundance of several factors involved in translation and mRNA decay in the *CFTR*-G542X 16HBEge cell line as well as in a promoter-enhanced *CFTR*-G542X 16HBEge cell line, which is a genetically modified version of the *CFTR*-G542X 16HBEge cell line that has increased CFTR expression via the stable integration of dCas9-VPR and a gRNA that enhances *CFTR* promoter activity and increases CFTR expression by 2–3-fold[42].

We performed western blotting on extracts from both cell lines to examine the abundance of the termination factors eRF1 and eRF3; translation factor eIF5A; ribosomal proteins RPL5 and RPL12; and the NMD factors SMG6 and UPF2 (Fig. 6a). Surprisingly, eRF1 abundance was reduced by ~70% following

SRI-41315 treatment (5 μM) but was unaffected by treatment with vehicle (0.1% DMSO) or G418 (100 μM) (Fig. 6a, b). To evaluate the possibility that SRI-41315 decreased antibody avidity to eRF1, we tested two different antibodies targeting the N- and

C-termini of eRF1 and obtained similar results (Fig. 6a). In contrast, the level of eRF3, eIF5A, RPL5, RPL12, and the housekeeping proteins vinculin and GAPDH were unaltered by SRI-41315 (Fig. 6a and Suppl. Fig. 7b-c).

We also measured eRF1 and eRF3 levels after SRI-41315 treatment alone and in combination with G418 in primary HBE cells (genotype W1282X/W1282X), as well as in promoter-enhanced *CFTR*-G542X 16HBEge cells[42] using the Simple Western technique. Again, we found similar reductions in eRF1 steady-state levels (Suppl. Fig. 7d-e), while eRF3 levels remained largely unchanged. Interestingly, the abundance of *eRF1* mRNA (relative to *GAPDH*) changed minimally in promoter-enhanced *CFTR*-G542X 16HBEge cells treated with SRI-41315 (Suppl. Fig. 7f), despite the dramatic decrease in eRF1 protein, indicating that SRI-41315 acts on eRF1 at the post-transcriptional level.

Based on the fact that eRF3 is known to undergo ubiquitin-mediated proteasomal degradation[43], we hypothesized that SRI-41315 may act on eRF1 by a similar mechanism. To test this hypothesis, we treated promoter-enhanced *CFTR*-G542X 16HBEge cells with the proteosome inhibitor (S)-MG132. We found that SRI-41315-mediated eRF1 degradation was prevented by the addition of (S)-MG132 but not the neddylation inhibitor MLN4924 (Fig. 6c). Similarly, readthrough stimulated by SRI-41315 in 16HBE14o- cells stably expressing the dual RT/NMD W134X NanoLuc reporter was reduced by (S)-MG132 under

**Table 1 Physiochemical properties of SRI-37240 and SRI-41315.**

| Assay | Target values | SRI-37240 | SRI-41315 |
|---|---|---|---|
| MW (g/mol) | <500 | 387.5 | 357.4 |
| HLM t1/2 (min) | >60 | 82 | 300 |
| MLM t1/2 (min) | >30 | 10 | 205 |
| Solubility (μM) | >10 | <1 | 51 |
| Log D | 2–4 | 4.5 | 2.4 |
| Polar surface area | 40–90 | 53 | 53 |
| Cell permeability | $10 \times 10^{-6}$ cm/s, ER<2 | $7.3 \times 10^{-6}$, ER = 6 | $22.99 \times 10^{-6}$, ER = 1.08 |
| CC50, FRT cells | >50 μM | >50 μM | >50 μM |
| CC50, 16HBE cells | >50 μM | >50 μM | >50 μM |

CC50 values (50% cytotoxic concentration) are defined as the compound's concentration (μg/mL) required for the reduction of cell viability by 50%.

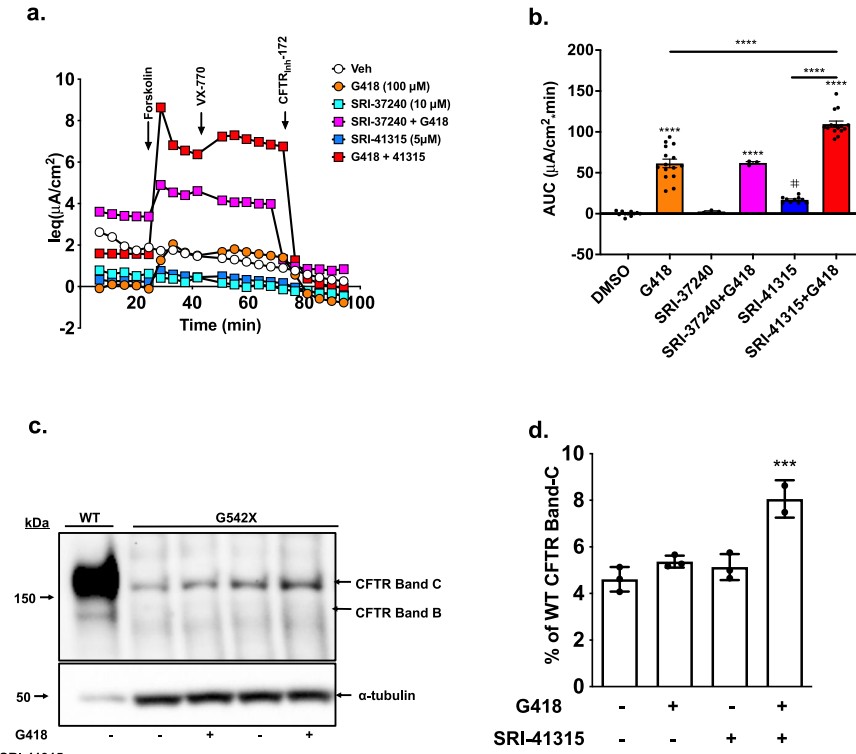

**Fig. 5 Combining G418 with SRI-41315 enhanced CFTR expression and function in *CFTR*-G542X 16HBEge cells.** *CFTR*-G542X 16HBEge cells were treated with SRI-37240 (10 μM) or SRI-41315 (5 μM) alone and in combination with G418 (100 μM) for 72 h. CFTR-specific channel activity in treated cells was compared to a vehicle control (DMSO) in a liquid–liquid interface. **a** Representative equivalent current ($I_{eq}$) tracings for SRI-37240 and SRI-41315 treatments alone and in combination with G418 vs. the DMSO control. **b** Corresponding summary data calculated as area under curve (AUC) between forskolin-induced (10 μM) stimulation of CFTR activity and inhibition of CFTR with CFTR$_{inh}$-172 (20 μM) ($n = 3$-15, data are expressed as mean ± S.D. and statistically analyzed using ordinary one-way ANOVA followed by Tukey's post hoc test, #$p = 0.0541$ (Vehicle vs SRI-37240), ****$p < 0.0001$). **c** Representative western blot showing the levels of full-length CFTR band C (fully processed) and B (unprocessed) after combination treatment compared to single-agent treatments. The total protein loaded was 3 μg for wild-type samples and 10 μg for G542X samples. **d** Quantitation of CFTR western blotting data (normalized to tubulin) ($n = 3$, data are expressed as mean ± S.D. and statistically analyzed using ordinary one-way ANOVA followed by Tukey's post-hoc test, ***$p = 0.0008$). Each experiment was performed 4–5 times with $n = 3$-4 monolayers per condition on each repeat. Source data is available as a source data file.

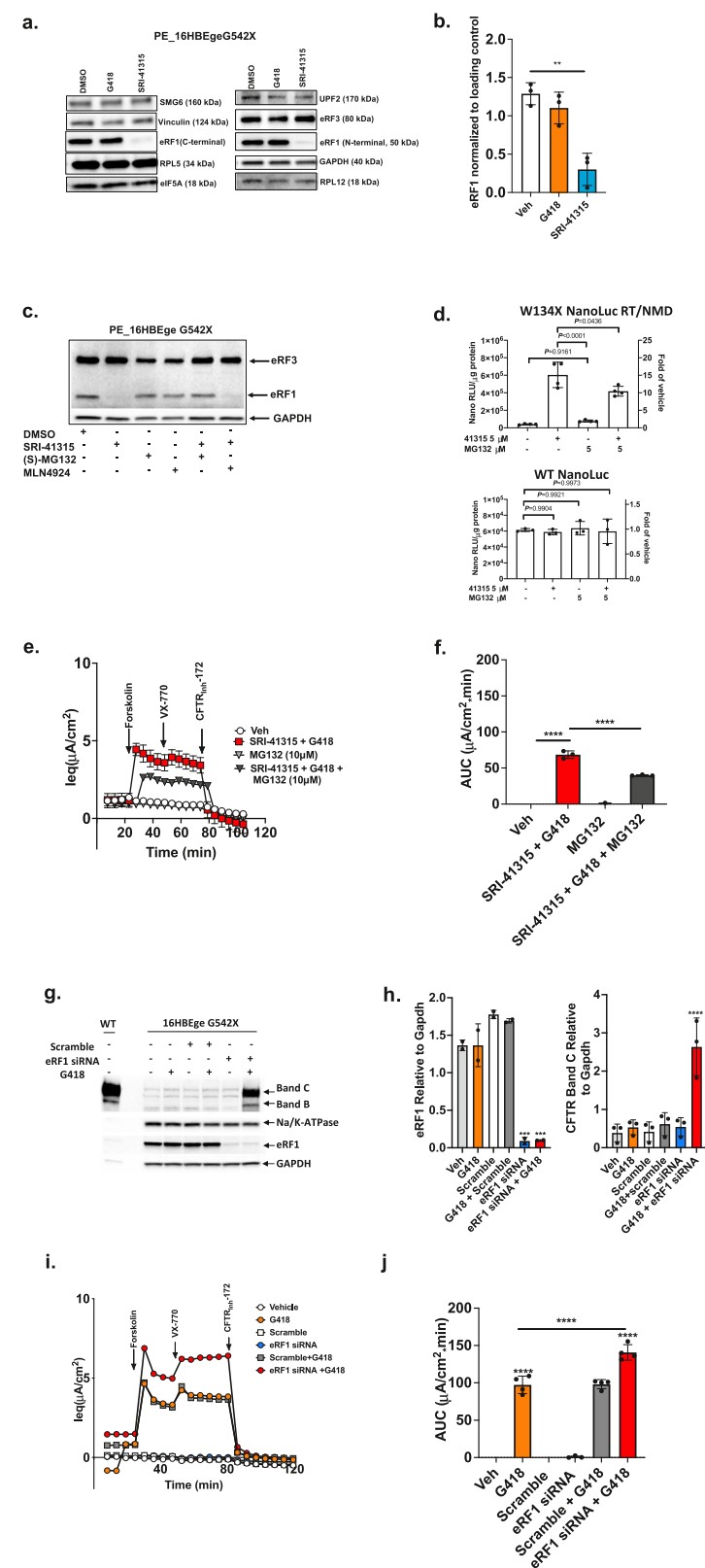

conditions where wild-type NanoLuc activity was not affected (Fig. 6d). Finally, functional rescue of CFTR in *CFTR*-G542X 16HBEge cells treated with G418 (100 μM) + SRI-41315 (5 μM) for 12 h was significantly diminished when (S)-MG132 (10 μM) was co-administered (Fig. 6e, f; Suppl. Fig. 1g shows that the

TEER values were comparable). To confirm that eRF1 reduction was at least partially responsible for enhanced readthrough in combination with G418, we reduced eRF1 expression using siRNA-mediated knockdown in *CFTR*-G542X 16HBEge cells (Fig. 6h; Suppl. Fig. 7h). eRF1 knockdown alone did not increase

**Fig. 6 SRI-41315 depletes eRF1 levels through a proteasome-mediated degradation pathway.** Promoter-enhanced *CFTR*-G542X 16HBEge G542X cells were grown on plastic and then treated with SRI-41315 (5 μM), G418 (100 μM), or DMSO control for 48 h. **a** Representative western blot showing protein expression levels of the translation termination factors eRF1 and eRF3, the translation factor eIF5A, the ribosomal proteins RPL5 and RPL12, and the NMD factor proteins SMG6 and UPF2 relative to controls vinculin (A Left) or GAPDH (A Right). **b** Western blot quantification of eRF1 protein relative to vinculin after treatment with G418, SRI-41315, or vehicle ($n = 3$, data are expressed as mean ± S.D. and statistically analyzed using ordinary one-way ANOVA followed by Dunnett's post hoc test, **$p = 0.0012$). **c** Representative western blot from cells after 20 h treatment with SRI-41315 (5 μM) alone and/or in combination with the proteasome inhibitors (S)-MG132 (10 μM) or MLN4924 (10 μM). **d** NanoLuc activity (normalized to total cellular protein) in 16HBEσ⁻ cells stably expressing the RT/NMD W134X or wild-type NanoLuc reporters and treated with SRI-41315 (5 μM) or MG132 (5 μM) for 24 h or MG132 (5 μM) for 12 h alone or combined. Data are expressed as the mean$+/-$SD. Exact $p$ values comparing cohorts (as indicated by the brackets) were determined using two-way ANOVA Multiple Comparisons ($n = 4$ for the RT/NMD reporter and $n = 3$ for the WT reporter). This assay was independently repeated with similar results. **e** CFTR activity was measured in *CFTR*-G542X 16HBEge cells treated with MG132(10 μM), SRI-41315 (5 μM) + G418 (100 μM), and in triple combination with SRI-41315 + MG132 + G418 for 12 h. Shown are average (±S.E.M.) equivalent current ($I_{eq}$) tracings from $N = 3$ replicates; **f** the corresponding summary data calculated as area under curve (AUC) between forskolin-induced (10 μM) onset of CFTR activity and inhibition with CFTR$_{Inh}$-172 (20 μM) ($n = 3$, data are expressed as mean ± S.D. and statistically analyzed using ordinary one-way ANOVA followed by Tukey's post hoc test, ****$p < 0.0001$). **g** Representative western blots of eRF1 and CFTR bands B and C relative to GAPDH and Na/K-ATPase in *CFTR*-G542X 16HBEge cells after eRF1 siRNA-mediated knockdowns alone (96 h), in combination with G418 (total 72 h with G418 replenishment at 48 h). **h** Corresponding quantification of westerns for eRF1 (left) and CFTR band C (right) relative to GAPDH in *CFTR*-G542X 16HBEge cells ($n = 3$, data are expressed as mean ± S.D. and statistically analyzed using ordinary one-way ANOVA followed by Dunnett's post hoc test, eRF1 Relative to Gapdh: ***$p = 0.0002$ (Vehicle vs eRF1 siRNA), ***$p = 0.0002$ (Vehicle vs eRF1 siRNA+G418); CFTR Band C Relative to Gapdh: ****$p < 0.0001$ (Vehicle vs eRF1 siRNA+G418). **i** Representative equivalent current ($I_{eq}$) tracings corresponding to CFTR activity in *CFTR*-G542X 16HBEge cells after siRNA-mediated knockdown of eRF1 alone, in combination with G418, or with scramble control treatments. **j** Corresponding $I_{eq}$ summary data calculated as area under curve (AUC) between forskolin (10 μM) and ivacaftor induced onset of CFTR activity and inhibition with CFTR$_{Inh}$-172 (20 μM) ($n = 4$, data are expressed as mean ± S.D. and statistically analyzed using ordinary one-way ANOVA followed by Tukey's post hoc test, ****$p < 0.0001$). Each experiment was performed 3–4 times with $n = 3$–4 monolayers or wells from 96-well plates per condition on each repeat. For all panels, ***$p \le 0.001$, ****$p \le 0.0001$. Source data is available as a source data file.

CFTR band C (Fig. 6g, h) or CFTR function (Fig. 6i, j; Suppl. Fig. 7i), but both were significantly increased when eRF1 knockdown was combined with G418 as compared to G418 alone (Fig. 6g–j). Taken together, these results indicate that SRI-41315 promotes readthrough and augments CFTR function by diminishing eRF1 protein abundance through a proteasomal degradation-dependent pathway.

**SRI-41315 in combination with G418 restores CFTR function in primary human bronchial epithelial cells homozygous for nonsense mutations.** We next asked whether CFTR activity can be restored in primary HBE cells expressing endogenous PTCs, since these cells have been used to predict clinical efficacy for CFTR modulators[44,45]. We used cells from a donor heterozygous for the *CFTR* R553X/W1282X mutations to observe the ability of SRI-37240 and SRI-41315 to rescue CFTR-mediated Cl⁻ transport activity using the TECC $I_{eq}$ assay (Fig. 7a–c). Administered alone, neither SRI-37240 (10 μM) nor SRI-41315 (5 μM) increase conductance significantly as compared to cells treated with vehicle alone (Fig. 7a, b), while G418 (100 μM) alone only produced a modest ~3-fold increase in conductance compared to the control. When SRI-41315 was combined with G418 (100 μM), a small, but significant increase in forskolin-stimulated current was observed ($I_{eq}$, 1.06 ± 0.38 AUC/min (μA/cm²) compared to either SRI-41315 (0.12 ± 0.02 AUC/min (μA/cm²) or G418 (0.30 ± 0.02 AUC/min (μA/cm²) alone. Interestingly, SRI-37240 also demonstrated modest efficacy in primary cells in combination with G418 (0.48 ± 0.09 AUC/min (μA/cm²), albeit with lower efficacy than SRI-41315. Unfortunately, SRI-37240 and SRI-41315 also had unanticipated and deleterious effects on ion conductance mediated by channels other than CFTR that was independent of readthrough, limiting their developmental potential in their current form as a treatment for CF (Supplementary Info.; Suppl. Fig. 8). Nevertheless, these results firmly establish the potential for developing small molecules that promote readthrough by the depletion of eRF1.

## Discussion

We embarked on a high throughput screen to identify novel small molecules that promote readthrough of PTCs, with the ultimate goal of identifying agents with novel mechanisms of action and increased efficacy. To do this, we developed a readthrough reporter that utilizes NanoLuc luciferase activity as its functional readout. NanoLuc is an engineered luciferase originally isolated from the deep-sea shrimp *Oplophorus*[46] that has no homology to other luciferases. NanoLuc is a small, 19 kDa monomeric protein that is structurally and thermally stable and produces a glow-type luminescence, which makes it an effective reporter for HTS. It also utilizes the novel engineered substrate, furimazine, making it less likely to generate false-positive responses that have been observed with other luciferases such as firefly luciferase[47,48]. It also has high sensitivity and a broad dynamic range. Based on these characteristics, the dual RT/NMD NanoLuc reporter described here represents a useful tool for HTS of small molecule libraries to identify compounds that stimulate readthrough, inhibit NMD (or increase mRNA abundance via other mechanisms), or target both processes to maximize overall protein function.

The effectiveness of this reporter for high throughput screening and subsequent hit validation was demonstrated by the identification and characterization of SRI-37240, a compound that both stimulates readthrough and increases mRNA abundance, leading to synergistic increases in protein function that make it a promising readthrough compound with a mechanism of action that has not been previously reported for a pharmacological agent. Ribosome profiling revealed a global increase in ribosome density at NTCs in cells treated with SRI-37240, suggesting that it acts by inhibiting translation termination. We also found that SRI-41315 leads to a depletion of eRF1, resulting in a prolonged pause at stop codons. Notably, ribosome profiling also indicated that SRI-37240 did not promote suppression of NTCs throughout the transcriptome. This important finding suggests SRI-37240, unlike G418, possesses considerable selectivity for readthrough of PTCs over normal termination codons. This is a clear advantage for a readthrough agent destined for clinical use.

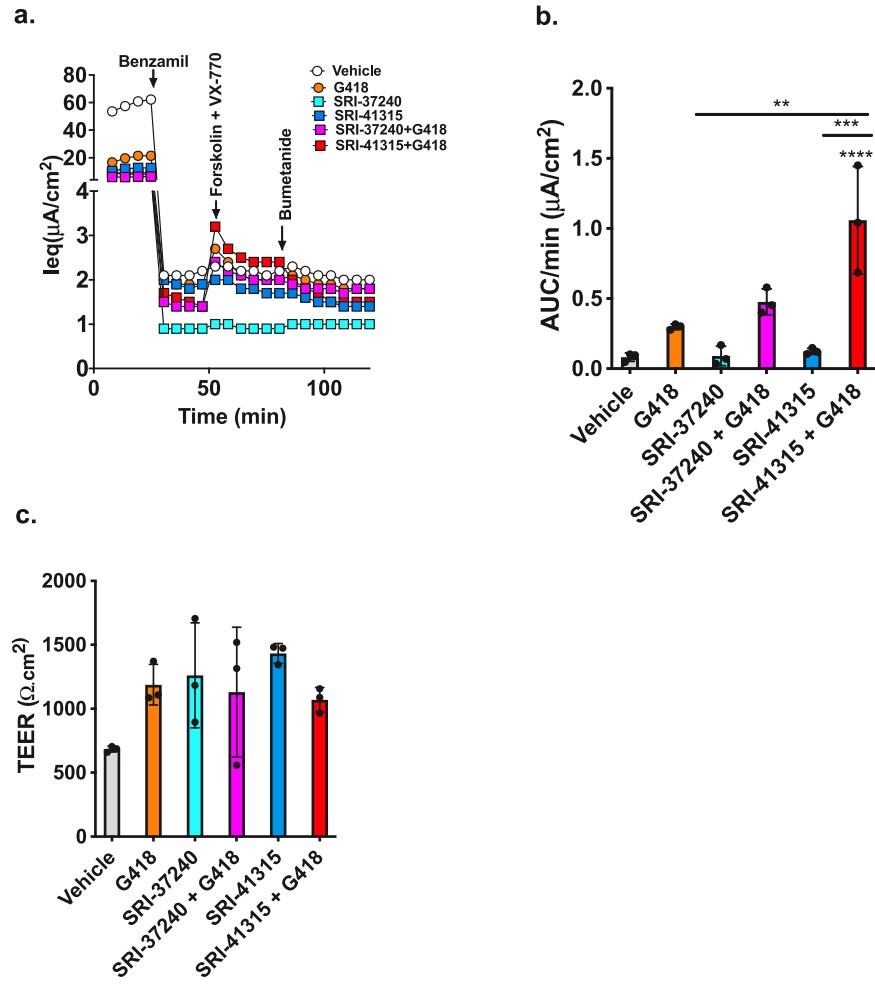

**Fig. 7 CFTR function is restored by SRI-37240 and SRI-41315 in combination with G418 in primary bronchial epithelial cells derived from a CF patient with *CFTR* nonsense alleles.** Primary HBE cells derived from a donor with an R553X/W1282X *CFTR* genotype were grown at an air–liquid interface until terminally differentiated and then treated with SRI-37240 (10 μM) or SRI-41315 (5 μM) alone or in combination with G418 (100 μM) for 72 h and then CFTR function was the measured. **a** Representative $I_{eq}$ traces after acute treatments with benzamil (10 μM), forskolin (10 μM) + ivacaftor (1 μM), and bumetanide (20 μM). **b** Corresponding average current calculated from the area under the curve between forskolin/ivacaftor-induced CFTR activity onset and anion transport inhibition with bumetanide ($n = 3$, data are expressed as mean ± S.D. and statistically analyzed using ordinary one-way ANOVA followed by Tukey's post hoc test, \*\*$p = 0.0010$ (G418 vs SRI-41315 + G418), \*\*\*$p = 0.0001$ (SRI-41315 vs SRI-41315 + G418), \*\*\*\*$p < 0.0001$ (Vehicle vs SRI-41315 + G418). **c** Transepithelial electrical resistance (TEER) values for the treatment conditions in panels (**a**) and (**b**) prior to electrophysiologic agonists ($n = 3$, data are expressed as mean ± S.D. and statistically analyzed using ordinary one-way ANOVA (P = NS)); for all panels, \*\*$p \leq 0.01$, \*\*\*$p \leq 0.001$, \*\*\*\*$p \leq 0.0001$. Source data is available as a source data file.

When FRT cells were evaluated for CFTR-dependent chloride conductance, we found that SRI-37240 exhibited readthrough activity that was synergistic with sub-optimal concentrations of G418, suggesting that it functions via a mechanism that is distinct from aminoglycosides. The magnitude of CFTR function restored by SRI-37240 in FRT cells was also significant with other *CFTR* nonsense mutations (R553X, R1162X, and W1282X) (Suppl. Fig. 4), indicating the general efficacy of this readthrough compound for multiple *CFTR* alleles. While total activity was most pronounced with the W1282X allele, it appears that residual CFTR activity was associated with the truncated peptide, a finding reported previously[15,49], which could be augmented with either nonsense suppression or CFTR modulators (Suppl. Fig. 4). Notably, these responses from other alleles were also relatively large with respect to isogenic wild-type CFTR expressing cells, ranging from 2 to 14%. It is likely that the heterologous cDNA expression in FRT cells results in an overestimation of CFTR activity with respect to the proportion of wild-type currents expected in the context of the native gene that will produce transcripts that are more likely to be subject to NMD.

In agreement with this, we found that in the 16HBEge cell lines harboring genomic *CFTR* nonsense mutations, SRI-37240-mediated readthrough was detectable only when it was combined with G418.

We synthesized and evaluated over 40 chemical derivatives of SRI-37240 to identify SRI-41315, a compound that exhibited better ADME properties and improved readthrough efficacy (Table 1). The improved activity of SRI-41315 was demonstrated in FRT cells using both NanoLuc reporter assays and cAMP-activated CFTR chloride conductance. These improvements also translated to human cells both alone and in combination with G418, noting that improved efficacy in primary HBE cells did require the latter (Fig. 7a–c). These findings clearly indicate that improvements in readthrough efficacy and pharmacologic properties of SRI-41315 were sufficient to enable translation to human airway cells that natively express *CFTR* transcripts with PTCs that are subject to NMD. Of note, while SRI-37240 was found in an unbiased fashion in our study, this compound, as well as several derivatives including SRI-41315, were previously described as readthrough agents in a Novartis patent[50].

Mechanistic studies further showed that the abundance of eRF1, a factor that recognizes all stop codons and is essential for translation termination[51], was diminished by SRI-41315 (Fig. 6a, b) and this effect was directly responsible for its effects on translational readthrough and rescue of CFTR (Fig. 7). This is consistent with a prior report that demonstrated enhanced readthrough following eRF1 depletion by small interfering RNAs (siRNA) or antisense oligonucleotides (ASOs)[51]. Molecular knockdown of eRF1 mRNA by ASOs was also recently shown to augment translational readthrough in hFIX-R338X hemophilia mice and was suggested as a robust target for inducing readthrough[52] and could be applicable to a wide range of other genetic diseases[53].

We further showed that SRI-41315 reduced eRF1 levels in a manner dependent upon a ubiquitin-mediated proteasome degradation pathway. A preliminary structural analysis did not find an obvious binding site on eRF1 for these molecules. This is not surprising, since our compounds may act in a manner analogous to the small molecule CC-885 that targets "neo-substrates" such as Ikaros and GSPT1 to the E3 ubiquitin ligase cereblon[54]. Neither of those proteins have appreciable affinity for the E3 ligase in the absence of this small molecule.

Taken together, these data suggest that SRI-41315 depletes eRF1, which induces a prolonged translational pause at PTCs that leads to near-cognate amino acid insertion, but without inducing significant readthrough of normal stop codons. To our knowledge, this is the first example of a pharmacological agent that can induce readthrough by affecting eRF1 levels, demonstrating an important therapeutic target for nonsense mutation suppression. While SRI-37240 and SRI-41315 had undesirable effects on sodium transport by ENaC that limit their immediate development for the treatment of CF, they may be useful in other disease models that are caused by PTCs. In general, their safety with regard to cellular toxicity, their pharmacological properties, and the general safety of their mechanism as suggested by ASO-mediated knockdown[52] suggest that approaches that reduce eRF1 abundance may serve as a potential therapeutic approach for PTC suppression.

Although aminoglycosides have demonstrated translational readthrough in vitro and been confirmed in preliminary proof-of-concept studies in humans[20,21,55], neither amikacin nor gentamicin rescues enough CFTR function to reach the therapeutic threshold necessary to impart long-term clinical benefits for CF. Furthermore, their toxicity and lack of oral bioavailability provide severe challenges for long-term administration[56]. The clinical development of the aminoglycoside derivative ELX-02, formerly NB124, could represent an alternative aminoglycoside with which eRF1 inhibition might be combined, given that ELX-02 is optimized for mammalian readthrough on cytosolic ribosomes while limiting effects on mitochondrial ribosomes, and has activity that approximates G418 in many PTC contexts[57]. Although we cannot explain why SRI-37240 was highly efficacious in FRT cells but only moderately active in 16BHEge and HBE cells when combined with G418, it could be because of differences in cellular physiology or proteostasis, subtle differences in the abundance of translation termination factors in rats vs. human cells, pharmacokinetics, or other properties that influence intracellular compound availability. These findings suggest that the use of human cells as an initial screening modality may be superior for the discovery of efficacious readthrough agents, contrasting with our initial assumption that the high conservation of the translation machinery among mammalian species would allow cross utility and consistent efficacy among a range of mammalian cell types.

In this study, we report the first pharmacological agent known to reduce eRF1 protein abundance, a mechanism well-suited to augment readthrough. While interventions that only target eRF1 may not be sufficient to provide a therapeutic benefit for many genetic diseases, the success of multi-drug combinations for CFTR modulator therapy for individuals with a F508del allele has established that addressing the molecular defects of CFTR protein misfolding, trafficking and activation can have a profound impact on overall CFTR activity. It has become increasingly likely that multiple distinct agents with different mechanisms of action will be required to impart a clinically impactful response[7,8,58]. Likewise, translation termination at PTCs is a complex process that occurs in multiple steps, providing several independent ways to reduce termination efficiency that include prolonging the translational pause that occurs during PTC recognition[59], depleting the translation termination factors required to complete translation[52], increasing the avidity of near-cognate tRNA binding[60], and inhibiting nonsense-mediated mRNA decay[61]. While further medicinal chemistry is needed to identify readthrough compounds that maximally impact CFTR function without undesirable off-target effects, the results presented here suggest this path is clearly achievable.

## Methods

**Generation of NanoLuc RT/NMD reporter cell line.** A Nanoluc-based reporter to identify compounds that induce readthrough (RT) and/or inhibit NMD was generated based on the wild-type (WT) NanoLuc gene that originated from plasmid pFN(Nluc_CMV_Neo) (Amp) (Promega). A multi-cloning site was placed into the XhoI/NotI sites of the parental plasmid using annealed oligos DB4078 and DB4079 (all oligos are located in Table 2 of the Supplementary Information) in order to remove the barnase sequence and to assist with subsequent cloning steps. This new construct (pDB1333) was used as a template to introduce the W134X UGA mutation (followed by A) by site-directed mutagenesis with primers DB4144 and DB4145 (Suppl. Table 2) to generate pDB1345. The NheI/NotI fragment of the W134X NanoLuc gene from pDB1345 was subcloned into pcDNA3.1Zeo(-) to generate W134X NanoLuc/pcDNA3.1Zeo(-) (pDB1357). The Renilla gene within the Renilla-WT β-globin/pcDNA3.1Zeo(-) (pDB1329) was replaced with the NheI/XhoI fragment of pDB1357 to fuse the W134X NanoLuc gene with exon 1 of the β-globin gene and generate the W134X NanoLuc RT/NMD reporter (pDB1368). This construct was verified by sequencing. This RT/NMD NanoLuc plasmid was transfected into FRT or 16HBE cells using Lipofectamine™ 2000 (Invitrogen-ThermoFisher, Cat#11668-019). After transfection, cells stably expressing the reporter were selected with Gibco™ Zeocin™ Selection Reagent (Fisher Scientific, Cat#R25005) for 2–3 weeks to generate polyclonal stable cell lines and then maintained in media containing 200 μg/mL zeocin for FRT cells and 40 μg/mL for 16HBE cells. Monoclonal stable lines were further established by expanding single-cell clones.

**NanoLuc assays.** Reporter cell lines were seeded into 96-well plates without zeocin for 24 h or until the cells reached 50–60% confluency. The cells were then treated with readthrough or DMSO vehicle (Sigma-Aldrich, Cat#D2650) for 24–48 h. Cells were then washed once with Corning™ Cell Culture Phosphate Buffered Saline (PBS) (1X) (Fisher Scientific, Cat#MT21040CV). Passive lysis buffer (PLB; 20 μL) (Promega, Cat#E194A) was added to each well to lyse the cells. Cell lysate (5 μL) from each well was used to perform the NanoLuc assay with the Nano-Glo® luciferase assay system (Promega, Cat#N1120) using the GloMax® Discover System (Promega). The lysate protein concentration was measured with the Bio-Rad Protein Assay Dye Reagent Concentrate (Bio-Rad, Cat#5000006). NanoLuc luminescence was normalized to total protein in each lysate and expressed as RLU/μg protein. The data represents the mean ± SD of 3–4 replicates.

**High throughput screening.** A collection of 771,345 unique, non-proprietary compounds custom assembled at Southern Research from various commercial vendors (ChemBridge, Enamine, ChemDiV, Life Sciences, Tripos) was screened in a HTS campaign using FRT cells that stably express the W134X NanoLuc RT/NMD reporter. The compounds were selected to have molecular properties matching eight criteria for lead-like molecules to serve as starting points for a drug discovery effort (molecular weight < =500; heteroatom count < =10; number rotatable bonds < = 8; number aromatic rings < =4; ALog P < = 6; molecular polar surface area < =200; H-bond acceptors < 10; H-bond donors < 5). The collection is diverse, containing over 37,000 clusters (Tanimoto distance <0.2) and over 275,000 non-overlapping Murcko scaffolds (determined by locating all ring systems of the molecule and all direct connections between them), including over 9000 individual ring systems and ~16,000 contiguous ring systems with unique substitution pattern.

SRI-37240 was one of the compounds in the Enamine library (T6740530). Notably, we were not aware of the previous patent from Novartis regarding the readthrough properties of this compound at the time of the screen. Based on the structure of SRI-37240 compound as part of a medicinal chemistry optimization,

SRI-41315 was synthesized at Southern Research and was not part of the original screening collection. Stock concentrations of library compounds (at 5 mg/mL or 10 mM dissolved in 100% DMSO) were screened at 30 μg/mL or 60 μM in the presence of 0.6% DMSO. Cells grown in BioChrom Coon's F12 Medium (Cedar Lane Labs Catalog # F 0855) supplemented with 5% heat-inactivated fetal bovine serum (Omega Scientific Catalog #FB-11) and 1% L-glutamine, were harvested and re-suspended at 400,000 cells/mL in media supplemented with 1% Pen/Strep/Glut (Corning Catalog # 30-009CI) and 1% HEPES (Gibco Catalog # 15630-080). Using a Wellmate, 25 μL of cells (10,000 cells/well) were added to all wells of Corning 384-well white opaque plates (Catalog #3570BC) to which 5 μL of compounds or controls at 6X final concentration in media had been previously added using a Beckman Biomek FX. Columns 1 & 2 of each plate contained media + 0.6% DMSO as the negative (cell only) control. Columns 23 & 24 contained 300 μg/mL of G418 Sulfate (Corning Catalog # 30-234-CI) + 0.6% DMSO as the positive control. The plates were gently tapped and incubated for 48 h at 37 °C with 5% $CO_2$. Following incubation, the plates were equilibrated to room temperature and room temperature NanoGlo (Promega Catalog # N1150) was added to all wells. After 10 min the luminescence of each well was measured using a PerkinElmer Envision plate reader. Active compounds resulted in an increased luminescent signal in the assay. The raw luminescence data for the test compounds were normalized relative to the mean luminescence of the 32 positive control wells (300 μg/mL G418) on each plate and reported as % Activation with 100% Activation being the mean of the positive control signals according to the following calculation: % Activation = 100*(Cmpd−Neg Ctrl)/(Pos Ctrl−Neg Ctrl).

**Ribosome profiling**. HEK293T cells (ATCC) were cultured in 10-cm dishes containing Dulbecco's modified Eagle medium with high glucose, L-glutamine, and sodium pyruvate, supplemented with 10% fetal bovine serum (Thermo Fisher Scientific) and grown in a 37 °C incubator with 5% $CO_2$. Cells were seeded in triplicate and treated for 24 h with DMSO (0.1%, Millipore Sigma), G418 (0.5 mg/mL, Thermo Fisher Scientific), or SRI-37240 (10 μM). Lysates were generated by dislodging cells by scrapping, centrifuging cells at room temperature for 5 min at $400 \times g$, aspirating media, and lysing cells for 10 min in 200 μL of ice-cold lysis buffer (20 mM Tris-Cl pH 7.4, 150 mM NaCl, 5 mM $MgCl_2$, 1 mM DTT, 100 μg/mL cycloheximide, 1% Triton X-100, 2.5 U/mL Turbo DNase I (Thermo Fisher Scientific). Lysates were then clarified by centrifugation for 10 min at 4 °C and $21,000 \times g$. The soluble fraction was quantified using the Quant-iT RiboGreen RNA Assay.

Sequencing libraries were prepared as previously described[62] with several minor modifications[36]. Briefly, 20 μg of RNA was digested with 750 U of RNase I (Ambion) for 1 h at 25 °C and stopped with 10 μL of SUPERase*In (Thermo Fisher Scientific). Ribosomes were pelleted through a 1 M sucrose cushion at 100,000 RPM, 4 °C, for 1 h in a TLA 100.3 rotor using a Beckmann-Coulter Optima MAX ultracentrifuge. Ribosome-protected fragments (RPFs) were extracted using the miRNeasy Mini Kit (Qiagen), and 15–35 nucleotide fragments were size-selected by denaturing PAGE using a 15% TBE-Urea gel. RPFs were dephosphorylated using T4 PNK (New England Biolabs), ligated to a 3′ oligonucleotide adapter with a unique molecular identifier (UMI) hexanucleotide degenerate sequence for 3 h at 37 °C using T4 RNA ligase 2 – truncated (New England Biolabs), and depleted of rRNA using RiboZero (Illumina) omitting the final 50 °C incubation step. RNA was then reverse-transcribed using Superscript III (New England Biolabs) using an RT primer with a second four-nucleotide UMI sequence, RNNN. cDNA was circularized with circLigase (Lucigen), amplified by PCR with Phusion high-fidelity polymerase (New England Biolabs), and quantified using a Bioanalyzer 2100 (Agilent). Illumina sequencing was performed on a NovaSeq at the Johns Hopkins Institute of Genetic Medicine. Data analysis was performed as previously described[36]. Samples were analyzed individually (3 replicates for each condition) and also after merging replicates. Results were similar between replicates, so all data presented here represent merged samples for simplicity. For preprocessing of raw sequencing files, UMI's were removed from the 5′ end with seqtk[63], adapters were trimmed from the 3′ end using skewer[64], noncoding RNA sequences were subtracted with STAR[65], and remaining reads were mapped to the hg38 annotation of the human genome guided by annotation files from GENCODE (version 30)[66]. All further data analysis was performed using custom Python 2.7 scripts available at https://github.com/jrw24. The Ribosome ReadThrough Score (RRTS) was defined as the density of ribosomes in the 3′-UTR between the NTC and the first in-frame 3′-termination codon divided this value by the density of ribosomes in the CDS for every annotated transcript (as previously described[36]). Stop codon pause scores were determined for each transcript by dividing the density of ribosomes at the stop codon by the normalized ribosome densities for each coding sequence, excluding the first 15 and last 12 nucleotides.

**Cell culture**. Fischer Rat Thyroid (FRT) cells: Isogenic FRT cells were stably transduced using the Flp-In™ System (Thermo Fisher Scientific, Inc.) with either a wild-type human *CFTR* cDNA or with a human *CFTR* cDNA harboring a clinically relevant *CFTR* nonsense mutation (G542X, R553X, R1162X, and W1282X) under *CMV* promoter control. Cell lines were expanded in cell culture flasks and then seeded on Transwell® Permeable Supports (cat no. 3378, Corning) in complete growth media until electrophysiologic assays were conducted, as described previously[16].

**Gene-edited human bronchial epithelial (HBE) cells**. The original 16HBE14o-cell line was gene-edited to introduce a G542X nonsense mutation into the endogenous *CFTR* allele (CFF-16HBEge CFTR G542X, referred to as 16HBEge G542X in this manuscript) and thus, is expressed via *CFTR* promoter control[40]. The cells were characterized as described previously[40,67]; their genome was shown to have at least two copies of the CFTR locus, with one copy being non-functional due to insertion of an SV40 sequence and inactivation of the synthesized mRNA. We also utilized a version of this cell that was further genetically modified by stable integration of dCas9-VPR and gRNA to enhance activity of the CFTR promoter[42] (thereafter referred to as 'promoter enhanced' 16HBEge G542X), which increases CFTR expression by 2–3-fold.

**Primary human bronchial epithelial (HBE) cells**. After expansion of P1 cells in BronchiaLife Airway Epithelial Cell Growth Medium (#LL-0023, Lifeline), P2/P3 primary HBE cells expressing *CFTR* R553X/W1282X (gift from Philip Karp, Univ. of Iowa) were seeded at a density of $5 \times 10^5$ cells/cm$^2$ (170,000 cells per filter) on HTS Transwell 24-well filter inserts (cat. #3378, Corning). Cells were differentiated at an air–liquid interface (ALI) for 4–5 weeks before use in equivalent current ($I_{eq}$) studies as described below.

**Cytotoxicity assay**. Cytotoxicity was assessed by using Cell Titer-Glo (Promega) to measure ATP levels as an index of cell viability. Compounds were diluted in assay media to prepare a 6x concentrated dosing solution and added to 384-well black clear-bottom plates in 5 μL (1/6 final well volume) using a Beckman FX liquid handler. One hundred (100) μM hyamine (at 0.3% DMSO) served as the low signal control (0% viability) and cells (at 0.3% DMSO) as the high signal control (100% viability). Twenty-five (25) μL of cells were added to all wells for a final count of 10,000 cells/well to give a final assay volume of 30 μL and a final DMSO concentration of 0.3% in all wells. All plates were incubated at 37 °C/5% $CO_2$ for 3 days in a humidified atmosphere. After equilibration to room temperature, 30 μL of Cell Titer-Glo (Promega) was added to each well and the plates incubated at room temperature for 10 min. Luminescence was measured using a BMG PheraStar reader. The % cell viability was calculated as follows: % viability = 100*(test cmpd value − mean low signal control)/(mean high signal control − mean low signal control). CC50 values were calculated from a four-parameter logistic fit of data using the Xlfit module of ActivityBase.

**FRT transepithelial chloride conductance measurements ($G_t$)**. Transepithelial conductance ($G_t$) was measured using a 24-channel transepithelial current clamp (TECC-24) amplifier (EP Design BVBA) as previously described[68] for monolayers of FRT cells harboring *CFTR* G542X, R553X, R1162X, or W1282X nonsense mutations. Cells were then treated with readthrough compounds or vehicle control for 48 h and CFTR-mediated changes in Cl⁻ conductance were measured upon stimulation of CFTR with forskolin (10 μM), ivacaftor (formerly VX-770, 10 μM), and inhibition of CFTR by $CFTR_{Inh}$-172 (10 μM)[15].

**16HBEge transepithelial chloride equivalent current measurments ($I_{eq}$)**. 16HBEge cells were seeded on HTS 24-well Transwell® Permeable Supports (Corning #3378) at $0.15 \times 10^6$ cells/well. Cells were submerged for 7 days in MEM media (Gibco) with 10% FBS prior to the assay, with media refreshed every other day. Once ready for assay, 16HBEge cells were incubated with test compounds via the apical and basolateral compartments for 72 h, with fresh medium and compound supplied after 48 h. For equivalent current measurements, a basolateral to apical chloride ion gradient was imposed and the assay was performed, as described previously[40]. $I_{eq}$ changes were assessed after addition of the cAMP agonist forskolin (10 μM), ivacaftor (1 μM), and $CFTR_{Inh}$-172 (20 μM). The area under the curve (AUC) between addition of the CFTR agonist (forskolin) and CFTR inhibitor (45 min) was used to determine functional expression/rescue of CFTR; for these data, AUC is plotted; AUC/min can be determined by dividing by the 45-min duration of CFTR activation.

**Primary HBE transepithelial chloride equivalent current measurements ($I_{eq}$)**. Air–liquid interface cultures were treated with compounds for 72 h. $I_{eq}$ current measurements were performed as described previously[69]. In the recorded $I_{eq}$, the first and second 4 data points reflect baseline currents under conditions where the apical epithelial Na⁺ channel (ENaC) is active and inhibited (after benzamil addition; 10 μM). Subsequent data points reflect CFTR-mediated changes in transepithelial Cl⁻ current upon addition of forskolin, ivacaftor, and the Cl⁻ channel inhibitor, bumetanide (20 μM). The magnitude of these changes or the integral of CFTR-mediated current over time, the AUC, relative to vehicle treatment, is a measure of small molecule-mediated functional rescue of mutant CFTR; data are reported as AUC/min of CFTR activation.

**Primary HBE short-circuit current measurements ($I_{sc}$)**. Ussing chamber short-circuit current measurements were performed across differentiated primary HBE cultures exposed to an air–liquid interface for 4–5 weeks on Snapwell permeable filter supports (Corning, #3801) as described previously[6].

**Western blotting**. CFTR: FRT and 16HBEge cells were washed with PBS, harvested, and lysed with ice-cold RIPA buffer containing 1% EDTA and 1% protease inhibitors (Halt Thermoscientific protease inhibitor cocktail, Catalog # PI78425, Thermo Fisher). Protein concentrations were determined using the Pierce™ BCA Protein Assay Kit (Catalog # PI23250, Thermo Fisher). Monoclonal CFTR primary antibody (UNC 596; 1:5000; received under MTA from the University of North Carolina at Chapel Hill) and mouse monoclonal anti-α-tubulin (1:5000; Catalog # 62204, Thermo Fisher) antibodies were used to detect CFTR and α-tubulin levels, respectively. Protein bands were visualized using Thermo Scientific™ SuperSignal™ West Femto Chemiluminescent Substrate (Catalog # PI34096, Thermo Fisher) for CFTR proteins or Thermo Scientific™ SuperSignal™ West Pico PLUS Chemiluminescent Substrate (Catalog # PI34579, Thermo Fisher) for α-tubulin. Images were captured by a Gel Doc XR + Gel Documentation System (Bio-Rad).

**Translation factors**. Well-differentiated primary *CFTR* W1282X/W1282X HBE cells and 16HBEge cells grown on filter supports or on plastic were treated with compound for 48 h. Cells were lysed in Pierce™ IP Lysis Buffer (87788) with complete protease inhibitor cocktail (Roche, 11836153001). Lysates were separated on a 4–15% polyacrylamide gel (Mini-PROTEAN® TGX™ Precast Gels, Bio-Rad, #4561086), transferred onto a nitrocellulose membrane (Bio-Rad, #1620112), and blocked in 5% (w/v) nonfat milk in TBST. Membranes were incubated with 1:1000 dilutions of antibodies to the N-terminus of eRF1 (Cell Signaling #13916) the C-terminus of eRF1 (Santa Cruz (#sc-365686), eRF3 (Cell Signaling, #14980), RPL5 (Cell Signaling, #14568), RPL12 (Abcam, #ab157130), eIF5A (Cell Signaling, #20765), UPF2 (Cell Signaling, #11875), or SMG6 (Abcam # ab87539) in blocking buffer overnight at 4 ℃. After three 10-min washes, the membrane was incubated with anti-Rabbit IgG (H + L) HRP (Jackson ImmunoResearch, #111-035-144, 1:10,000 dilution) or anti-mouse IgG (H + L) (Jackson ImmunoResearch, #115-035-166, 1:10,000 dilution) in blocking buffer for 1 h at room temperature, followed by three wash cycles with TBST. Proteins were visualized using the Clarity™ Western ECL Substrate (Bio-Rad, #1705060). Images were acquired using the Gel Doc XR + Gel Documentation System (Bio-Rad). After protein signals were captured, membranes were washed with TBST three times for five minutes each to remove the ECL substrate. Membranes were then incubated with antibodies to either GAPDH (1:2000, Cell Signaling, #2118) or vinculin (1:2000, Cell Signaling, #13901) for 2 h. Steps for secondary antibodies incubation and signal capture were as described above.

eRF1 and eRF3 ProteinSimple western blotting: Promoter-enhanced 16HBEge G542X cells were seeded at $2 \times 10^5$ cells per well in a 24-well plate. After 24 h, the medium was replaced with fresh compound-containing medium and cells were incubated for 24–48 h. Cells were lysed in 100 μL Pierce™ IP Lysis Buffer (87788) with complete protease inhibitor cocktail (Roche, 11836153001). Capillary electrophoresis western blot analysis was carried out with a Wes (ProteinSimple) equipment and ProteinSimple reagents according to manufacturer protocols using default settings. Briefly, 4.8 μL of cell lysate was mixed with 1.2 μL of fluorescent master mix and heated at 95 ℃ for 5 min. The samples, blocking reagent, wash buffer, and antibodies for eRF1 (1:50, Cell Signaling, #13916), eRF3 (1:30, Cell Signaling, #14980), GAPDH (1:1000 or 1:2000, Cell Signaling, #2118), and vinculin (1:2000 or 1:5000, Cell Signaling, #13901), as well as secondary antibodies and chemiluminescent substrates, were loaded into the ProteinSimple kit microplate. The data were analyzed with Compass software (ProteinSimple).

**RNA silencing**. ETF1 (eRF1) siRNA (J-019840-05-0050) and a scrambled control siRNA (D-001206-13-05) were purchased from Horizon Discovery. Transfections were performed using Lipofectamine RNAiMAX transfection reagent (ThermoFisher, 13778150) according to manufacturer's protocol. Briefly, 16HBEge G542X cells ($4.52 \times 10^5$ cells/cm²) were seeded onto Corning HTS Transwell-24 plates (Corning, 3378) coated with collagen IV (Sigma, C5533-5MG) and grown in culture medium without antibiotics a day before transfection. siRNAs (30 ηM) were applied on both the apical and basolateral sides of the membrane overnight in antibiotic-free medium. Then, transfection medium was then replaced and G418 (100 μM) was applied to both apical and basolateral sides of related wells at day 3 and day 5 post-transfection. TECC assays were performed on day 6 post-transfection.

**Quantitative real-time PCR (qRT-PCR)**. RNAeasy isolation kit (Qiagen) was used to isolate total RNA. RNA concentration was quantitated using a NanoDrop (Thermofisher). Quantitative real-time PCR was carried out using TaqMan One-Step PCR master mix[70] using primers purchased from Life Tech (Suppl. Table 2). *CFTR* and *eRF1* transcript levels were quantified relative to *GAPDH* or *TBP*[15,16,70].

**In vitro pharmacology assessments**. The distribution coefficient (LogD) was measured after compounds were partitioned between n-octanol and phosphate-buffered saline (PBS, pH 7.4). Kinetic solubility was measured at pH 7.4 after 10 mM solutions of compound (in DMSO) were added to PBS buffer. The residual DMSO in PBS was 1%. LC-MS method was used to quantify compounds in LogD and solubility studies.

Metabolic stability of compounds was evaluated in human and mouse liver microsomes. The final concentration of study compound was 1 μM, microsome protein concentration was 0.5 mg/mL in 50 mM potassium phosphate buffer at pH

7.4, with 1.0 mM NADPH and 3.3 mM MgCl₂. The disappearance of parent compounds was measured by LC-MS/MS. CACO-2 permeability studies were performed using CacoReady kit (ADMEcell), between 21 and 25 days after seeding. The CACO-2 monolayer integrity was assessed by transepithelial resistance (TEER) measurements prior to each experiment. The transport of compounds (10 μM) was examined in the apical-to-basolateral (AB) and basolateral-to-apical (BA) directions in HBSS buffer (with 10 mM HEPES, pH 7.4) in a cell culture incubator (37 °C with 5% CO₂, 95% relative humidity). Samples were analyzed by LC-MS/MS. The apparent permeability ($P_{app}$) and efflux ratios (ER) were calculated using the following equations:

$$P_{app} = \frac{\Delta Q}{\Delta t \times A \times C0} \tag{1}$$

$$ER = \frac{P_{app}(B-A)}{P_{app}(A-B)} \tag{2}$$

Where $\Delta Q/\Delta t$ is the amount of translocated material over incubation time; $A$ is the area of transwell; and C0 is the initial concentration.

**In vitro transcription/translation assay**. The TnT® SP6 Coupled Reticulocyte Lysate System (Promega, cat#: L4600) was used to generate the protein products expressed from the QXQ readthrough plasmid (pDB603)[41] which contains a UGA(C) PTC sequence. A 25 kDa protein was produced by efficient termination at the PTC, while a 35 kDa protein was generated by readthrough of the PTC. The Transcend™ tRNA (Promega, cat.# L5061) was added to the system for the chemiluminescent detection of the protein products via incorporation of biotinylated lysines. The assay was performed according to manufacturer's instructions. The readthrough agents G418 or SRI-41315 were added to the reactions at 0.2 and 0.4 μg/ml, or at 5 and 10 μM, respectively. The reactions were incubated at 30 °C for 90 min and then 2 μL from each reaction was loaded onto precast 4–15% polyacrylamide gels (Bio-Rad, cat#4568084) to separate proteins via SDS-PAGE electrophoresis. The proteins were then transferred to Immobilon-P membrane-FL (Millipore), which was incubated with the Streptavidin-HRP conjugate (1:10,000 dilution) for 60 min, washed, and then incubated with SuperSignal™ West Femto Maximum Sensitivity Chemiluminescent HRP substrate (ThermoFisher Scientific cat#34094) for 1 min. A Chemi-Doc™ Imager (Bio-Rad) was used to detect the signal.

**Statistics**. Statistical analysis was carried out using GraphPad Prism v5 software (GraphPad Software Inc.) and *p*-values <0.05 were considered statistically significant. For assays with multiple conditions, two-way ANOVA or one-way ANOVA followed by Tukey's post hoc testing for multiple comparisons tests were used to conduct inferential statistics between treatment conditions. Statistics are presented as (mean ± SD) SEM, unless indicated otherwise. All data points were used unless otherwise stated. Dropped data points are indicated in the Source data file.

**Reporting summary**. Further information on research design is available in the Nature Research Reporting Summary linked to this article.

## Data availability
All relevant data are available from the authors. The raw data from each figure or table is available in the Source data file. For the ribosomal profiling data, the raw sequencing data and count tables for each sample were deposited in the Gene Expression Omnibus database (GSE144140). Source data are provided with this paper.

## Code availability
Analysis of ribosome profiling data was performed using custom Python 2.7 scripts available at https://github.com/jrw24/SRI37240. Source data are provided with this paper.

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

## Acknowledgements

This study was supported by the NIH (to S.M.R., D.M.B., and E.J.S.) and the Cystic Fibrosis Foundation.

## Author contributions

C.E.A., R.J.B., V.M., M.M., K.M.K., R.G., D.M.B., and S.M.R. conceived of the experiments. J.S., M.D., Y.L., J.C., J.W., L.P.T., E.W., F.L., and H.B. conducted the research. D.M.B., K.M.K., M.M., A.R., J.H., E.M.M., E.J.S., C.E.A., R. G., M.M., and S.M.R. provided new reagents and techniques. J.S., Y.L., M.D., J.W., R.G., M.M., C.E.A., D.M.B., and S.M.R. analyzed the data. J.S., K.M.K., S.M.R., and D.M.B. wrote the manuscript. M.J.S., D.M.B., and S.M.R. supervised the project. All authors had an opportunity to edit the manuscript and approved of its submission.

## Competing interests

This work was supported by funding from NIH and the CF Foundation. M.M., E.W., F.L., and H.B. are the employees of the Cystic Fibrosis Foundation. D.M.B. is a paid consultant for PTC Therapeutics, Inc. and Zikani Therapeutics, Inc. S.M.R. reports grants from PTC Therapeutics, grants, personal fees and non-financial support from Vertex Pharmaceuticals Incorporated, grants, personal fees and non-financial support from Novartis, grants from Eloxx, and grants from Galapagos/Abbvie during the conduct of the study. Grants from Bayer, grants from Forest Research Institute, grants from AstraZeneca, grants from N30/Nivalis, grants from Galapagos/AbbVie, grants from Proteostasis, grants and personal fees from Celtaxsys, personal fees from Bayer, personal fees from Renovion, grants and personal fees from Synden/Synspira, personal fees from Genentech, personal fees and non-financial support from Boeringher Ingelheim, grants from Jannssen, Vivus, Actelion, Johnson and Johnson, and other related entities are outside the submitted work. All other authors declare no competing interests.
