## [Peer Review File · Nature Communications]

Reviewers' Comments:

Reviewer #1:

Remarks to the Author:

The manuscript from DU, M et al. aims at identifying a new readthrough-inducing compound that could be used to restore CFTR activity. They have developed three/four reporter systems and performed an HTS that allowed them to identify one compound named SRI-37240. This compound displays an interesting suppression efficiency and seems to restore CFTR activity from the Y122X mutation, in an artificial rat system (FRT cell line). SRI-37240 has already been described in a patent in 2014 as a readthrough inducer (this is not mentioned in this manuscript or I missed the information). In the corresponding patent (WO2014091446) the effect of the compound is tested using the CFTR Y122X mutation. So data provided here are only very partially new, and unfortunately there is no mechanistic information about the action of the compound that would add some novelty.

You will find below my comments/questions to the authors:

- In the material and methods, it is indicated that constructs are stably integrated and then that clones are selected. On which criteria?
- The first four figures are dedicated to characterization of three different reporter systems that are not even for the screening. This is rather surprising. Why the authors did not evaluate the reporter they used for the HTS? This one carries a new stop mutation, and there is no information about its ability to monitor NMD or readthrough.
- I am not entirely convinced by the interest of having developed so many different reporter systems, that seems to monitor imperfectly especially the NMD pathway. Figure 1C, how the authors explain the appearance of the truncated form of the NanoLuc with G418 in a system that is supposed to be insensitive to NMD? Moreover, the controls used for KD experiments seems to have a clear and significant impact on the readout (Figures 2D, 3C and 3D).
- Figure 2. The interest of the Q39X mutation in the globin part is not clear for me, while the natural NanoLuc stop should elicit NMD by itself due to the presence of downstream EJC junctions?
- Figure 2. How confident the authors are that this system accurately monitors NMD? UPF2KD has a very limited (if any) effect on the Nanoluc activity (x1.5) while a 4-fold change in mRNA level is observed. The latter result is also observed with a Ctrl KD in presence of G418 (Figure 2D) that should not impact NMD at all. Authors proposed that G418 could induce a change in the mRNA localization or in translation efficiency, but these hypotheses are not supported by data. My conclusion is that this system monitors NMD very imperfectly.
- P8 (Figure 2C and 2D): does the effect also observed with the reporter without the stop codon at the end of the NanoLuc gene?
- Figure 5. The HTS has been performed with another reporter system (W134X) that has never been tested neither for NMD nor readthrough.
- Figure 5B, C, D, E. Why changing the representation of the data? This is very confusing and makes comparison difficult. If I understood correctly SRI-37240 is 7x more active than G418 on the dual RT/NMD system (Figure 5B), but almost 2x less active than G418 on the RT reporter system (Figure 5C) and shows a 1.5-fold increase with the NMD reporter (Figure 5D). How a compound that is 2x less active on readthrough and stabilizes 1.5-fold the mRNA can show a 7-fold increased on the dual reporter? I think this perfectly reflects the issues with the various reporters used in this work.
- Figure 6. The RiboSeq data interestingly show that ribosomes accumulate to the stop codons without noticeable effect on their ability to readthrough in presence of SRI-37240. It is actually surprising that ribosomes pausing at the stop codons are not prone to readthrough more efficiently. This is even more surprising if we take into account that SRI-37240 induces eRF1 degradation, this must impact all stop codons in the cells not only PTC.
- P14, (Figure 6). It is unclear to me if the signal (ribosome density) downstream the stop codon corresponds only to the reads that are in-frame with the previous CDS or to all reads whatever is the frame? This must be clarified.
- P17, there is no indication about SRI-37240 toxicity.

- Figure 7. It would be helpful to have the Δ isc trace in presence of WT CFTR to be able to compare the levels obtained with the drugs to the WT protein. FRT is a heterologous system (Rat cell line) used to quantify CFTR activity. Bedwell's and Rowe's groups have access to primary cells that would be far more relevant in a therapeutic perspective.
- P18. The authors claim that SRI-37240 is the first compound possessing a clear selectivity for readthrough of PTCs over NTCs. This is not true as the same finding has been reported for Ataluren (PTC124).
- References: I noticed a very strong tendency for self-citation. Actually 16 out of 47 references are from the authors' groups (34%).

Reviewer #2:

Remarks to the Author:

Identifying compounds that mediate efficient readthrough and at the same time inhibit NMD would be ideal for nonsense suppression therapy. Du and co-workers describe here their development and testing of different variants of NanoLuc reporter constructs that allow to identify and distinguish readthrough and NMD inhibition (Figs. 1-4). Subsequently, they used the dual NanoLuc RT/NMD reporter for a high throughput screen, in which they identified 180 compounds that suppressed the premature termination codon (PTC) in the reporter. The manuscript finishes with the characterization of one of the top hits in the screen, SRI-37240. This compound causes efficient PTC readthrough and a modest increase of the mRNA level (presumably due to NMD inhibition) (Fig. 5). Ribosome profiling showed that SRI-37240 globally increases the ribosome density at the stop codon without globally enhancing readthrough, opposite to G418, which globally enhances readthrough and leads to a slightly reduced ribosome density at the stop codon (Fig. 6). Finally, they authors used a cellular cystic fibrosis model that relies on a CFTR minigene with the Y122X mutation and could show that SRI-37240 alone can restore more CFTR activity than any of the previously developed drugs (including G418) and when combined with a SMG1 inhibitor and ORKAMBI (an FDA-approved drug comprising VX-809 and VX-770), 15% of the wild-type CFTR activity could be restored. These results identify SRI-37240 as a promising readthrough agent for follow-up studies exploring its potential for nonsense suppression therapy of various genetic diseases that are caused by nonsense mutations. Overall, the manuscript also makes an important point about the therapeutic potential of combining several readthrough promoting and NMD inhibiting compounds, as several combinations clearly show strong synergistic effects.

Altogether, I find the work highly interesting, succinctly written and the data for the most part compelling and therefore support publication of this manuscript after the following points have been satisfactorily addressed:

Major points:

- The NanoLuc dual RT/NMD reporter seems to be much more sensitive to report readthrough (20-fold up with 300 ug/ml G418) than NMD (2 to 3-fold up under UPF2 KD). The apparently weak response to NMD however might be to moderate NMD inhibition under the achieved UPF2 reduction. The authors should therefore test alternative approaches to reach maximal NMD inhibition, in order gain more accurate information about this reporter's sensitivity to NMD. For example, the SMG1 inhibitor used in Fig. 7 could be combined with UPF1 or UPF2 KD, or Cycloheximide could be tried.
- It seems unlikely that the faint bands observed with G418/UPF2KD/(Amlexanox) treatment represent full-length NanoLuc protein, since even under efficient readthrough conditions, there would still be much more truncated than full-length NanoLuc protein be produced (see Fig. 1C). Since the gel has run in a skewed way, judging the size of that bands is difficult and they might actually represent truncated NanoLuc protein. The authors should either unambiguously identify these bands or delete panel 3E. If it is confirmed that these bands indeed represent full-length NanoLuc protein, how would the authors explain that no truncated NanoLuc protein can be detected on this blot?

- More detailed information should be given about the small compound library/libraries used for the screen. Where does it come from? How has it been composed? Are these proprietary compounds and if so, has SRI-37240 been patented? If this is the case, the patent should be referenced. Furthermore, I am also missing more information about the results of the screen. Minimally, a list of the 180 positive hits should be provided in supplementary materials or the link to the entire raw data of the screen deposited in a public repository should be provided.
- The Discussion focuses much on the specificity of SRI-37240 to promote readthrough selectively at PTCs but not at normal stop codons. However, the authors do not show much evidence for this. Does the ribosome profiling reveal increased readthrough at known endogenous NMD-sensitive transcripts compared to NMD sensitive ones? Or does treatment of cells with SRI-37240 lead to a specific upregulation of the RNA levels of endogenous NMD targets? Such data would be needed to corroborate the selectivity claims. Otherwise, in the absence of more compelling data on this issue, these claims should be toned down in the Discussion.

Minor points:

- Abstract, line 7: delete "with"
- References should be consecutively numbered. The mix-up seems to have occurred because the Methods section was moved to the end of the paper without adjusting the numbering of the references accordingly.
- Fig. 1C: the tubulin control is useless and also confusing, since this is from a different gel on which different amounts of cell lysate was loaded. As a proper control, the membrane probed for NanoLuc should be re probed with Tubulin antibody.
- P.10: The 7-fold higher mRNA level observed when G418 was combined with the Ctrl KD compared to Ctrl KD alone is explained as "suggesting that the transfection conditions may have resulted in some degree of mRNA stabilization". This doesn't make sense: The transfection conditions are identical, the mRNA increase is instead caused by G418 addition and could be due to readthrough-mediated NMD inhibition or an NMD-independent mRNA stabilization caused by G418 (as was observed also in Figs. 1E and 2D).
- What was the reason for switching from the R154X to the W134X variant of the reporter for conducting the screen?
- Fig. 5A: As a reference point, the authors should indicate the average % activation reached with the 32 positive controls in this screen.
- Suppl. Fig 5A: Please clarify what the yellow boxes depict. What is the origin of these "intron consensus sequence"?

Reviewer #3:

Remarks to the Author:

This manuscript describes three new luciferase reporter assays for study of nonsense mutation suppression, a very large-scale screen for small molecules that suppress nonsense mutations, as well as the discovery of the active compound SRI-37240 and characterization of its activity in a cell model of cystic fibrosis.

The study is technically sound and identifies an interesting small molecule suppressor of nonsense mutations. However, it is a frustrating read as the paper lacks cohesion.

The first reporter is designed to monitor premature stop codon readthrough. It is not highly innovative as other similar assay have previously been published. A good characterization of this assay using the readthrough aminoglycoside G418 is presented in Figure 1.

The second reporter is designed to report on nonsense-mediated decay (NMD) and is characterized to some extent in Figure 2. It is also not very different from previously published work.

The third is a dual reporter designed to simultaneously report on both readthrough and NMD and it

incorporates features of the first and second reporters. This reporter is characterized in quite a bit of detail in two different cell lines in Figures 3 and 4.

Up to this point, the presentation of these three reporters represents a logical progression.

The next section presents a very large-scale high-throughput screen of small molecules to identify compounds that suppress nonsense mutations by inducing premature stop codon readthrough and/or by inhibiting NMD. The issue here is that the screen is not carried out with the dual reporter characterized in detail in the manuscript, but with a different dual reporter for which no characterization is reported. This reporter may be "a variant" of the extensively characterized reporter, as the authors state, but it is not the same. This is where the parts don't fit. Why devote two figures to the characterization of a reporter that is not used in the study and then present the key results (Figure 5) using a reporter that is not characterized in this paper?

Nevertheless, Figure 5 presents an impressive high-throughput screen of >700,000 compounds that revealed SRI-37240 as a nonsense suppressor.

The manuscript then goes on to characterize the effects of SRI-37240 by ribosome profiling. Aligning the data on the transcripts' stop codons revealed that SRI-37240 does not appear to induce readthrough of normal stop codons and that it causes ribosomes to pause at normal stop codons, by a mechanism that is not pursued in this paper. This is an interesting experiment that indicates that SRI-37240 acts by a mechanism that is distinct from that of the well characterized readthrough aminoglycoside G418.

The last part of the study shows restoration of CFTR activity determined in cells expressing a CFTR nonsense mutation reporter, by measuring the change in short circuit current after addition of forskolin, presented in Figure 7. This reviewer does not have the expertise to assess these functional data.

Major comments:

1- As described above, the use of a reporter in the high-throughput screen that is not the one characterized in Figures 3 and 4 is an important concern. Why not show characterization of the reporter that is used to discover SRI-37240?

2- SRI-37240 has previously been reported as being active in a "CFTR-Y122X" nonsense suppression assay in a 2014 Novartis patent (example 1.32, WO 2014/091446A1). This study may have rediscovered this compound independently. Nevertheless, it would seem appropriate to cite this previous discovery in the discussion.

3- Important information that is lacking in this paper is data on the cytotoxicity of SRI-37240 and of the different combination treatments shown throughout this paper. Cytotoxicity is an important concern to researchers in the nonsense suppression field as G418 and related small molecules such as ELX-02 can induce high levels of readthrough but only at concentrations that are also very cytotoxic (at 300 µg/ml, G418 is quite highly toxic to a variety of cell lines). The luciferase data presented here are normalized to cell lysate protein concentration, which masks any cellular toxicity. It would seem important to show whether the restoration of NanoLuc and CFTR activity by SRI-37240 alone and in combination with the different compounds shown in this study takes place at concentrations that are highly toxic or not.

4- Western blots are used in Figure 1C to conclusively demonstrate formation of full-length NanoLuc, the readthrough product. This is an important piece of information that is lacking in Figure 7. The study on CFTR would be considerably strengthened if restoration of full-length protein could be shown.

5- The ribosome profiling experiment shown in Figure 6 seems a bit out of context and is not accompanied by comments on how the data showing increased pausing of ribosomes at normal stop codons might shed light on the mechanism of action of SRI-37240. It is only when reaching the Discussion section that the reader learns that SRI-37240 acts by inducing the degradation of the termination factor eRF1, as well as mention of an accompanying paper. This reviewer has not seen this accompanying paper. Nevertheless, the increased ribosome density at stop codons makes complete sense if the compound reduces the abundance of eRF1, whose function is to accelerate translation termination. Would it make more sense to instead put Figure 6 in the accompanying paper? Or even to combine the two papers?

Minor comments:

1- The characterization of full-length NanoLuc protein formation in the dual RT/NMD reporter presented in Figure 3C is not convincing. The full-length NanoLuc band is very faint and difficult to see. In Figure 1C, the authors used 50 times more protein for the nonsense reporter cells than the WT reporter cells to clearly show induction of full-length protein. However, they did not do that in Figure 3C. As a result, readthrough is unconvincing. Additionally, this panel lacks a control KD, an important one given the results shown in 3D indicating that transfection reagents "may have resulted in some degree of mRNA stabilization" (p. 10).

2- Page 7: "passive mRNA stabilization associated with G418-induced readthrough. Passive is a biased term. Distinct from EJC-dependent NMD?"

Response to reviewer comments for *Nature Communications* Submissions NCOMMS-20-12893 [Du et al. submission] and NCOMMS-20-10191 [Sharma et al. submission].

We wish to thank the reviewers for their thoughtful insights on these two studies. Overall, we believe the reviewers did an excellent job of identifying weaknesses in our initial submissions. We made extensive modifications of both manuscripts and have now merged these two manuscripts into one as requested, creating a more cohesive study. We hope these changes will address all reviewer concerns and comments.

To streamline and focus the manuscript, we omitted several figures that were least relevant to the combined dataset. Below, we present responses to each of the critiques raised for each of the two manuscripts. Many of the reviewer comments are no longer applicable since they were in reference to figures that were eliminated from the revised manuscript. We indicated this in our responses where appropriate, and specifically addressed all other reviewer concerns in more detail. Our responses are shown in red text. Because of the extensive rewrite necessitating the combination of the manuscripts, we have not included a track changes version.

Reviewers comments:

NCOMMS-20-12893

Reviewer 1.

The manuscript from DU, M et al. aims at identifying a new readthrough-inducing compound that could be used to restore CFTR activity. They have developed three/four reporter systems and performed an HTS that allowed them to identify one compound named SRI-37240. This compound displays an interesting suppression efficiency and seems to restore CFTR activity from the Y122X mutation, in an artificial rat system (FRT cell line). SRI-37240 has already been described in a patent in 2014 as a readthrough inducer (this is not mentioned in this manuscript or I missed the information). In the corresponding patent (WO2014091446) the effect of the compound is tested using the CFTR Y122X mutation. So data provided here are only very partially new, and unfortunately there is no mechanistic information about the action of the compound that would add some novelty.

We thank the reviewer for these comments. By combining our two manuscripts, the mechanism of action of SRI-37240 (and its derivative, SRI-41315), which is destabilization of eRF1 protein, is now shown. While there is information on the original compound SRI-37240 in the patent language, it is not easily accessible, has not been subject to peer review, and has not been published in the biomedical literature. SRI-41315, to our knowledge, is a novel agent.

You will find below my comments/questions to the authors:

- 1) In the material and methods, it is indicated that constructs are stably integrated and then that clones are selected. On which criteria?

Constructs were introduced on a plasmid with a Zeocin-resistant cassette. Zeocin was used to select stably transfected polyclonal cells and monoclonal isolates were then isolated from colonies and also maintained in the presence of Zeocin. Based on the basal reporter expression and the response to G418, the positive control readthrough

compound, a monoclonal cell line was selected to carry out the HTS. These details were added to the first section of the Materials and Methods "Generation of NanoLuc RT/NMD reporter cell line" on page 22.

- 2) The first four figures are dedicated to characterization of three different reporter systems that are not even for the screening. This is rather surprising. Why the authors did not evaluate the reporter they used for the HTS? This one carries a new stop mutation, and there is no information about its ability to monitor NMD or readthrough.

Sorry for the confusion. In the original submission, we wanted to describe the development of the entire group of reporters which were initially characterized in HEK293 cells. Later we decided to use FRT cell to do the HTS since FRT cells were widely used in CF research and have been used previously in successful screening campaigns. Since the W134X dual RT/NMD FRT clone #14 is highly responsive to G418 readthrough and NMD inhibition treatment, it was ultimately chosen for further HTS. Similar to the R154X dual RT/NMD reporters, the W134X RT/NMD reporter also synergistically responds to readthrough and NMD inhibition.

At the suggestion of some reviewers and the editor, we have now combined our two manuscripts describing SRI-37240 and its derivative SRI-41315 to make a more complete scientific report. As a result, the initial figures describing the family of reporters has been omitted due to lack of relevance. In the revised manuscript, we only use the W134X RT/NMD reporter, which clearly shows responses to readthrough compounds and NMD inhibitory agents (see Suppl. Fig. 1A-C).

- 3) I am not entirely convinced by the interest of having developed so many different reporter systems, that seems to monitor imperfectly especially the NMD pathway. Figure 1C, how the authors explain the appearance of the truncated form of the NanoLuc with G418 in a system that is supposed to be insensitive to NMD? Moreover, the controls used for KD experiments seems to have a clear and significant impact on the readout (Figures 2D, 3C and 3D).

As described above, we now only include the readthrough/NMD reporter used for the HTS. References to all other constructs were omitted from the revised manuscript.

- 4) Figure 2. The interest of the Q39X mutation in the globin part is not clear for me, while the natural NanoLuc stop should elicit NMD by itself due to the presence of downstream EJC junctions?

This construct was meant to use WT NanoLuc activity to report on decreased mRNA abundance due to NMD. Since this reporter is no longer included in this manuscript, this section has been removed.

- 5) Figure 2. How confident the authors are that this system accurately monitors NMD? UPF2KD has a very limited (if any) effect on the Nanoluc activity (x1.5) while a 4-fold change in mRNA level is observed. The latter result is also observed with a Ctrl KD in presence of G418 (Figure 2D) that should not impact NMD at all. Authors proposed that G418 could induce a change in the mRNA localization or in translation efficiency, but these hypotheses are not supported by data. My conclusion is that this system monitors NMD very imperfectly.

We agree that these reporters only partially reflect reductions in mRNA abundance due to NMD. However, this is no longer relevant to the revised manuscript since these data (and associated text) have been removed from the revised manuscript, allowing us to focus exclusively on the reporter used in the HTS screen.

- 6) P8 (Figure 2C and 2D): does the effect also observed with the reporter without the stop codon at the end of the NanoLuc gene?

Yes, they respond similarly. As described above, we have omitted our characterization of the family of reporters and included only information pertaining to the NanoLuc W134X RT reporter that was used for HTS. Thus, Figures 1-3 have been omitted.

- 7) Figure 5. The HTS has been performed with another reporter system (W134X) that has never been tested neither for NMD nor readthrough.

We include characterization of the NanoLuc W134X Readthrough/NMD reporter in Suppl. Fig. 1A-C of the revised manuscript.

- 8) Figure 5B, C, D, E. Why changing the representation of the data? This is very confusing and makes comparison difficult. If I understood correctly SRI-37240 is 7x more active than G418 on the dual RT/NMD system (Figure 5B), but almost 2x less active than G418 on the RT reporter system (Figure 5C) and shows a 1.5-fold increase with the NMD reporter (Figure 5D). How a compound that is 2x less active on readthrough and stabilizes 1.5-fold the mRNA can show a 7-fold increased on the dual reporter? I think this perfectly reflects the issues with the various reporters used in this work.

We have omitted all of the reporter data except for the W134X reporter that was used to perform the HTS. We have now maintained consistency throughout the combined manuscript in how we represented reporter activity to resolve the concerns noted by the reviewer.

- Figure 6. The RiboSeq data interestingly show that ribosomes accumulate to the stop codons without noticeable effect on their ability to readthrough in presence of SRI-37240. It is actually surprising that ribosomes pausing at the stop codons are not prone to readthrough more efficiently. This is even more surprising if we take into account that SRI-37240 induces eRF1 degradation, this must impact all stop codons in the cells not only PTC.

We think this interesting result provides evidence that the context of a stop codon and its proximity to factors bound to the 3'-UTR and/or poly(A) tail can influence its response to changes in eRF1 abundance. We comment on this point in the revised discussion.

- P14, (Figure 6). It is unclear to me if the signal (ribosome density) downstream the stop codon corresponds only to the reads that are in-frame with the previous CDS or to all reads whatever is the frame? This must be clarified.

The total ribosome density downstream of stop codons (independent of reading frame) was analyzed. To clarify this point, we added the statement "The Ribosome ReadThrough Score (RRTS) was defined as the density of ribosomes in the 3'-UTR between the NTC and the first in-frame 3'-termination codon divided by the density of ribosomes in the CDS for every annotated transcript" to the methods section on Ribosome Profiling.

- P17, there is no indication about SRI-37240 toxicity.

Table 1 has been included that summarizes the properties of SRI-37240 and SRI-41315, including CC50 (50% cytotoxic concentration) values.

- Figure 7. It would be helpful to have the Δ Isc trace in presence of WT CFTR to be able to compare the levels obtained with the drugs to the WT protein. FRT is a heterologous system (Rat cell line) used to quantify CFTR activity. Bedwell's and Rowe's groups have access to primary cells that would be far more relevant in a therapeutic perspective.

This control has been included in Suppl. Fig. 3. Primary HBEs are also utilized in Fig. 7.

- P18. The authors claim that SRI-37240 is the first compound possessing a clear selectivity for readthrough of PTCs over NTCs. This is not true as the same finding has been reported for Ataluren (PTC124).

We thank the reviewer for pointing this out. This statement has been removed.

- References: I noticed a very strong tendency for self-citation. Actually 16 out of 47 references are from the authors' groups (34%).

We have carefully examined the references that were cited. Where relevant, we have included the work of additional authors. Please note that many of the techniques/ approaches/reagents used in this study were pioneered by the co-authors of this study, which involve six separate laboratories. For example, Dr. Bedwell pioneered using nonsense suppression as a potential therapeutic approach for CF. Dr. Rowe led many of the clinical studies resulting in clinical use of CFTR modulators. Dr. Greene developed ribosomal profiling methodology to examine translation termination. Drs. Sorscher and Mense generated many of the CF-relevant cell lines. Since many of the co-authors are members of the CF research community who possess expertise in different areas, they routinely collaborate and co-publish their work together, which likely lends to the appearance of self-citation.

Reviewer 2.

Identifying compounds that mediate efficient readthrough and at the same time inhibit NMD would be ideal for nonsense suppression therapy. Du and co-workers describe here their development and testing of different variants of NanoLuc reporter constructs that allow to identify and distinguish readthrough and NMD inhibition (Figs. 1-4). Subsequently, they used the dual NanoLuc RT/NMD reporter for a high throughput screen, in which they identified 180 compounds that suppressed the premature termination codon (PTC) in the reporter. The manuscript finishes with the characterization of one of the top hits in the screen, SRI-37240. This compound causes efficient PTC readthrough and a modest increase of the mRNA level (presumably due to NMD inhibition) (Fig. 5). Ribosome profiling showed that SRI-37240 globally increases the ribosome density at the stop codon without globally enhancing readthrough, opposite to G418, which globally enhances readthrough and leads to a slightly reduced ribosome density at the stop codon (Fig. 6). Finally. They authors used a cellular cystic fibrosis

model that relies on a CFTR minigene with the Y122X mutation and could show that SRI-37240 alone can restore more CFTR activity than any of the previously developed drugs (including G418) and when combined with a SMG1 inhibitor and ORKAMBI (an FDA-approved drug comprising VX-809 and VX-770), 15% of the wild-type CFTR activity could be restored. These results identify SRI-37240 as a promising readthrough agent for follow-up studies exploring its potential for nonsense suppression therapy of various genetic diseases that are caused by nonsense mutations. Overall, the manuscript also makes an important point about the therapeutic potential of combining several readthrough promoting and NMD inhibiting compounds, as several combinations clearly show strong synergistic effects.

Altogether, I find the work highly interesting, succinctly written and the data for the most part compelling and therefore support publication of this manuscript after the following points have been satisfactorily addressed:

Major points:

- The NanoLuc dual RT/NMD reporter seems to be much more sensitive to report readthrough (20-fold up with 300 ug/ml G418) than NMD (2 to 3-fold up under UPF2 KD). The apparently weak response to NMD however might be to moderate NMD inhibition under the achieved UPF2 reduction. The authors should therefore test alternative approaches to reach maximal NMD inhibition, in order gain more accurate information about this reporter's sensitivity to NMD. For example, the SMG1 inhibitor used in Fig. 7 could be combined with UPF1 or UPF2 KD, or Cycloheximide could be tried.

Based on the recommendation of the editor, we combined our two manuscripts into one. As a result, it was prudent to omit the reporter studies with the exception of the W134X NanoLuc RT/NMD reporter that was used for HTS. In Suppl. Fig. 1A-C, characterization of this reporter shows that both NanoLuc activity and NanoLuc mRNA abundance are increased with UPF2 KD and G418 alone and when UPF2 KD and G418 treatments are combined, synergistic increases are observed in both NanoLuc activity and mRNA abundance. This clearly indicates that this reporter can be utilized to identify agents that induce RT or induce RT and alter mRNA abundance. The reporter dynamic range is broader for RT than for NMD, where the basal NanoLuc activity is $\leq 0.1\%$ of WT levels, while the mRNA abundance of the reporter is approximately 10-20% of WT levels. Thus, even if NMD is maximally inhibited, we would only expect a maximal 5-10-fold increase in mRNA abundance. Thus, the 2- to 3-fold increases observed with UPF2 KD are representative of the partial NMD inhibition that one might expect from small molecule inhibitors. By focusing on this single combined readthrough/NMD reporter, we present data most relevant to the HTS and also activity that is clearly induced by readthrough, mRNA abundance, or both mechanisms.

- It seems unlikely that the faint bands observed with G418/UPF2KD/(Amlexanox) treatment represent full-length NanoLuc protein, since even under efficient readthrough conditions, there would still be much more truncated than full-length NanoLuc protein be produced (see Fig. 1C). Since the gel has run in a skewed way, judging the size of that bands is difficult and they might actually represent truncated NanoLuc protein. The authors should either unambiguously identify these bands or delete panel 3E. If it is confirmed that these bands indeed represent full-length NanoLuc protein, how would the authors explain that no truncated NanoLuc protein can be detected on this blot?

This figure was deleted as suggested.

- More detailed information should be given about the small compound library/libraries used for the screen. Where does it come from? How has it been composed? Are these proprietary compounds and if so, has SRI-37240 been patented? If this is the case, the patent should be referenced. Furthermore, I am also missing more information about the results of the screen. Minimally, a list of the 180 positive hits should be provided in supplementary materials or the link to the entire raw data of the screen deposited in a public repository should be provided.

More details about the HTS are provided on page 6 of the Results. The library is a combination of proprietary and non-proprietary compounds. A patent has been filed for SRI-37240 and this is clearly mentioned on page 18 of the Discussion, the reference is cited, and the structures of both compounds are shown. Given the issue with preserving intellectual property for the other compounds, we prefer not to divulge the structures of these compounds as this would compromise their development as therapeutic agents, and they are not the focus of the current study or the mechanism revealed.

- The Discussion focuses much on the specificity of SRI-37240 to promote readthrough selectively at PTCs but not at normal stop codons. However, the authors do not show much evidence for this. Does the ribosome profiling reveal increased readthrough at known endogenous NMD-sensitive transcripts compared to NMD sensitive ones? Or does treatment of cells with SRI-37240 lead to a specific upregulation of the RNA levels of endogenous NMD targets? Such data would be needed to corroborate the selectivity claims. Otherwise, in the absence of more compelling data on this issue, these claims should be toned down in the Discussion.

The reporter and CFTR assays clearly demonstrated the ability of SRI-37240 and SRI-41315 to readthrough a PTC. In contrast, the ribosomal profiling examined ribosomal densities of global transcripts, where global endogenous transcripts were aligned at their natural stop codons. This assay showed that normal stop codons were not suppressed (as compared to the untreated negative control and the G418 positive control). These results form the basis of our comment. Dr. Green's lab has been working on the characterization of NMD substrates using ribosomal profiling for the last few years, and they still have not obtained data with adequate sequencing depth to be convincing. Thus, we hope the reviewer will agree that this request is beyond the scope of the current manuscript. We have revised the text in the Discussion regarding so that it speaks to the primary data in a way that does not overstate its effects.

Minor points:

- Abstract, line 7: delete "with"

Thanks, the abstract has been extensively revised.

- References should be consecutively numbered. The mix-up seems to have occurred because the Methods section was moved to the end of the paper without adjusting the numbering of the references accordingly.

Sorry for this mistake. The references have been revised and reformatted.

- Fig. 1C: the tubulin control is useless and also confusing, since this is from a different gel on which different amounts of cell lysate was loaded. As a proper control, the membrane probed for NanoLuc should be re probed with Tubulin antibody.

Sorry for the confusion and thanks for the suggestion. However, this figure was omitted when the manuscripts were combined.

- P.10: The 7-fold higher mRNA level observed when G418 was combined with the Ctrl KD compared to Ctrl KD alone is explained as “suggesting that the transfection conditions may have resulted in some degree of mRNA stabilization”. This doesn’t make sense: The transfection conditions are identical, the mRNA increase is instead caused by G418 addition and could be due to readthrough-mediated NMD inhibition or an NMD-independent mRNA stabilization caused by G418 (as was observed also in Figs. 1E and 2D).

This figure was omitted in the combined manuscript.

- What was the reason for switching from the R154X to the W134X variant of the reporter for conducting the screen?

In the combined manuscript we now only show data using the W134X reporter.

- Fig. 5A: As a reference point, the authors should indicate the average % activation reached with the 32 positive controls in this screen.

This is shown in Suppl. Table 1.

- Suppl. Fig 5A: Please clarify what the yellow boxes depict. What is the origin of these “intron consensus sequence”?

This figure (and experiment) was omitted in the revised manuscript.

Reviewer 3.

This manuscript describes three new luciferase reporter assays for study of nonsense mutation suppression, a very large-scale screen for small molecules that suppress nonsense mutations, as well as the discovery of the active compound SRI-37240 and characterization of its activity in a cell model of cystic fibrosis.

The study is technically sound and identifies an interesting small molecule suppressor of nonsense mutations. However, it is a frustrating read as the paper lacks cohesion.

The first reporter is designed to monitor premature stop codon readthrough. It is not highly innovative as other similar assay have previously been published. A good characterization of this assay using the readthrough aminoglycoside G418 is presented in Figure 1.

The second reporter is designed to report on nonsense-mediated decay (NMD) and is characterized to some extent in Figure 2. It is also not very different from previously published work.

The third is a dual reporter designed to simultaneously report on both readthrough and NMD and it incorporates features of the first and second reporters. This reporter is characterized in quite a bit of detail in two different cell lines in Figures 3 and 4.

Up to this point, the presentation of these three reporters represents a logical progression.

The next section presents a very large-scale high-throughput screen of small molecules to identify compounds that suppress nonsense mutations by inducing premature stop codon readthrough and/or by inhibiting NMD. The issue here is that the screen is not carried out with the dual reporter characterized in detail in the manuscript, but with a different dual reporter for which no characterization is reported. This reporter may be “a variant” of the extensively characterized reporter, as the authors state, but it is not the same. This is where the parts don't fit. Why devote two figures to the characterization of a reporter that is not used in the study and then present the key results (Figure 5) using a reporter that is not characterized in this paper?

Nevertheless, Figure 5 presents an impressive high-throughput screen of >700,000 compounds that revealed SRI-37240 as a nonsense suppressor.

The manuscript then goes on to characterize the effects of SRI-37240 by ribosome profiling. Aligning the data on the transcripts' stop codons revealed that SRI-37240 does not appear to induce readthrough of normal stop codons and that it causes ribosomes to pause at normal stop codons, by a mechanism that is not pursued in this paper. This is an interesting experiment that indicates that SRI-37240 acts by a mechanism that is distinct from that of the well characterized readthrough aminoglycoside G418.

The last part of the study shows restoration of CFTR activity determined in cells expressing a CFTR nonsense mutation reporter, by measuring the change in short circuit current after addition of forskolin, presented in Figure 7. This reviewer does not have the expertise to assess these functional data.

Major comments:

1- As described above, the use of a reporter in the high-throughput screen that is not the one characterized in Figures 3 and 4 is an important concern. Why not show characterization of the reporter that is used to discover SRI-37240?

We have now focused on the reporter that was used for the HTS and describe it exclusively in the revised manuscript.

2- SRI-37240 has previously been reported as being active in a “CFTR-Y122X” nonsense suppression assay in a 2014 Novartis patent (example 1.32, WO 2014/091446A1). This study may have rediscovered this compound independently. Nevertheless, it would seem appropriate to cite this previous discovery in the discussion.

We have cited this patent in the Discussion on page 18.

3- Important information that is lacking in this paper is data on the cytotoxicity of SRI-37240 and of the different combination treatments shown throughout this paper. Cytotoxicity is an important concern to researchers in the nonsense suppression field as G418 and related small molecules such as ELX-02 can induce high levels of readthrough but only at concentrations that are also very cytotoxic (at 300 µg/ml, G418 is quite highly toxic to a variety of cell lines). The luciferase data presented here are normalized to cell lysate protein concentration, which masks any cellular toxicity. It would seem important to show whether the restoration of NanoLuc and CFTR activity by SRI-37240 alone and in combination with the different compounds shown in this study takes place at concentrations that are highly toxic or not.

We have included Table 1 that includes toxicity information for both SRI-37240 and SRI-41315. The compounds are not toxic at the concentrations used.

4- Western blots are used in Figure 1C to conclusively demonstrate formation of full-length NanoLuc, the readthrough product. This is an important piece of information that is lacking in Figure 7. The study on CFTR would be considerably strengthened if restoration of full-length protein could be shown.

Although we have deleted the Figure 1C that includes western blotting of the NanoLuc reporter, we have included western blots showing full-length CFTR in Figures 2D and 5C.

5- The ribosome profiling experiment shown in Figure 6 seems a bit out of context and is not accompanied by comments on how the data showing increased pausing of ribosomes at normal stop codons might shed light on the mechanism of action of SRI-37240. It is only when reaching the Discussion section that the reader learns that SRI-37240 acts by inducing the degradation of the termination factor eRF1, as well as mention of an accompanying paper. This reviewer has not seen this accompanying paper. Nevertheless, the increased ribosome density at stop codons makes complete sense if the compound reduces the abundance of eRF1, whose function is to accelerate translation termination. Would it make more sense to instead put Figure 6 in the accompanying paper? Or even to combine the two papers?

Thanks for this comment as it reflects the problem with separate submissions describing the entire body of work. We have combined the two papers and now have experiments that directly show that a derivative of SRI-37240 induces eRF1 degradation.

Minor comments:

1- The characterization of full-length NanoLuc protein formation in the dual RT/NMD reporter presented in Figure 3C is not convincing. The full-length NanoLuc band is very faint and difficult to see. In Figure 1C, the authors used 50 times more protein for the nonsense reporter cells than the WT reporter cells to clearly show induction of full-length protein. However, they did not do that in Figure 3C. As a result, readthrough is unconvincing. Additionally, this panel lacks a control KD, an important one given the results shown in 3D indicating that transfection reagents “may have resulted in some degree of mRNA stabilization” (p. 10).

Due to combining our two manuscripts, this figure was omitted for lack of relevance.

2- Page 7: “passive mRNA stabilization associated with G418-induced readthrough. Passive is a biased term. Distinct from EJC-dependent NMD?”

Yes, we refer to a possible alternative mechanism that would be distinct from EJC-dependent NMD. By passive mRNA stabilization, we refer to a mechanism by which the stabilization of mRNA could depend on either altered translation rates and/or restored ribosome transit through the entire mRNA, which would be NMD-independent processes.

Specific comments on submission NCOMMS-20-10191 (Sharma et al.)

Reviewer 1.

This manuscript from Sharma J et al. aims at characterizing the SRI-37240 and one of its derivative SRI-41315 on their ability to suppress CFTR nonsense mutations. Moreover, authors indicate that these drugs induce the degradation of eRF1 (the protein that recognizes stop codons) explaining their action on stop codon readthrough. The manuscript is interesting although very marginally new in term of conceptual advance, the drug being previously described in a patent in 2014. Moreover, I am not entirely convinced by their data about the degradation of eRF1, as the same effect is although obtained with RPL12 and RPL5 (Figure 6A).

Finally, as the authors noticed, the drugs have already been described in a patent in 2014, and display undesirable effects on sodium transport, that limit their immediate development in therapeutics (P19, L421). So, these drugs will probably never be used in human medicine.

Response: We thank the reviewer for his/her generally favorable comments and useful critiques. In the revised manuscript, we now provide much more substantive data that definitively show that a reduction in eRF1 protein is responsible for the readthrough mechanism of the novel compound SR-41315, which has never been described in the scientific literature. We do agree with the reviewer that this drug itself cannot be used for the human use in its present form due to off-target effects. However, the manuscript does identify a first in class small molecule in an array of complementary systems, illustrating the potential utility of this mechanism for inducing translational readthrough, particularly when used as a combinatorial approach with aminoglycosides. It is our anticipation that this can be used to identify chemical matter with better drug properties that will facilitate human use, serving as a launch point for the development of human medicine of broad interest.

I have several other comments for the authors:

- Synergy and additivity are not the same. Throughout the manuscript authors should verify the use of these terms.

Response: This is an important point. We have assured our language regarding synergy vs. additivity is more precise in the revised manuscript.

- P6, L129: Comparison between SRI-37240 and G418 is not relevant (Figure 1A and 1B) because SRI-37240 was tested using the RT/NMD reporter and G418 with the RT reporter that does not take into account NMD. Both systems give different results that are difficult to compare. An accurate comparison would be SRI-37240 and G418 with the RT reporter.

Response: We thank the reviewer for this comment. As per the reviewer's suggestion and guidance from the editor, the newly combined manuscript focuses upon the single reporter system which is sensitive to both readthrough and NMD. This manuscript now shows the comparison between SRI-37240 and G418 in this reporter and resolves the reviewer's concern regarding challenge in interpretation of multiple different reporter constructs.

- P7, L145: I do not understand the rationale here. SRI-37240 stimulates 846-fold luciferase activity when it is dependent of both RT and NMD (Fig1A), and only 20fold when the reporter is insensitive to NMD. One may conclude that SRI-37240 mainly acts through NMD. However, this is not the authors' conclusion.

Response: This issue is resolved with use of the single dual-activity reporter presented in the revised manuscript.

- P9, L201. The observation that amikacin but not gentamicin has an effect with SRI-37240 is very interesting and could probably suggest differences in the mode of action. Is it also true with SRI-41315? If so, how is it compatible with the fact that SRI-41315 would act by promoting eRF1 degradation?

Response: We appreciate the reviewer for this comment and agree that differences in response among amikacin vs gentamicin in combination therapy is an interesting finding. We suspect the primary reason is that gentamicin is not as potent as amikacin, but differences in PK may also be a factor. There is no reason to suspect this is different between SR-41315 and SR-37240. As this is no longer the focus of the combined paper, this data has been omitted.

- P11, L228. Please specify the identity of the stop codon and the +1 nucleotide for all mutations tested.

Response: We thank the reviewer for this comment. We have updated the text with the identity and the +1-nucleotide information for all the mutations tested.

- P12, L263. SRI-37240 but not SRI-41315 is inactive in 16HBE14o- cells. What is the toxicity of the two compounds in the various cell lines used in this study?

Response: We provide cell cytotoxicity and other ADME profile data both SRI-37240 and SRI-41315 in Table 1. In addition, we also show transepithelial electrical resistance (TEER), a measure of cytotoxicity and cell monolayer integrity, from cells used during the assay. The data indicate cellular monolayers were intact at the dosage used in these studies and were not toxic to the cells when efficacy was ascertained. They further show that these properties are unlikely to distinguish SRI-37240 from SRI-41315.

- P14, L314. Quantification of the western-blot results would be very helpful to evaluate accurately the reduction of protein level. Although SRI-41315 has a clear effect on eRF1 amount, it also has an impact on RPL12, RPL15 levels and possibly on eIF5A.

Response: We agree that western blot quantification will help to clarify the distinct effect of SRI-41315 on eRF1. We repeated the blots with all the targets tested and included the quantification data in the revised manuscript. We did not see any significant impact on RPL5, RPL12 or eIF5. Please see the updated figures: Figure 6A, B and Supplementary Figure 7B-E.

- Do they authors' check the level of ribosomes in presence of SRI-41315? Do polysome profiles altered by the drug?

Response: We did not directly assess ribosome abundance or polysome profiles. However, to assess the effect of SRI-41315 on translation termination, we performed ribosomal profile using HEK293 cells treated with SRI-41315 or G418 (See Fig. 3). We found that for cells treated with G418, we saw a global decrease in ribosomal density at stop codons with a corresponding increase in ribosomal density bound to 3' UTRs,

indicating increased readthrough of stop codons globally. In contrast, for cells treated with SRI-41315, we found an increase of ribosomes bound to stop codons, indicating increased pausing at stop codons globally, but no increase in ribosomal density associated with 3' UTRs. These data indicate that G418 and SRI-41315 induce readthrough by different mechanisms.

- P15, L332. Does SRI-41315 work in vitro? This should be tested.

Response: To address this question, we tested SRI-41315 compared to G418 in a reticulocyte lysate coupled transcription/translation system to determine whether there were direct effects on translation. We found that G418, which targets the ribosome, induced readthrough and produced an increase in full-length protein. In contrast, we did not observe readthrough by SRI-41315 in this defined 'in vitro' assay, presumably because it could not act to reduce eRF1 levels in the translation extract, which lack the proteasome. This data corroborates with our findings that the readthrough effect generated by SRI-41315 is mediated through eRF1 depletion in a proteasome-dependent manner (see Supplementary Figure 7A).

General comments:

- How cells can survive with eRF1 being almost absent, this is an essential protein. So, if this is correct, one may expect a significant level of toxicity for SRI-41315. What about the initial drug (SRI-37240)? Does it also induce eRF1 degradation?

Response: We thank the reviewer for this comment. We repeated these western blots, and quantification showed about 70% reduction of eRF1 levels with SRI-41315 without any observed toxicity at the concentrations used. Nonetheless, recent study in haemophilia mice model (hFIX-R338X) using ASO mediated eRF1 knockdown has shown that ~60% inhibition of eRF1 transcripts and 40% of eRF1 protein levels in mouse liver safely induces readthrough without reducing body weights, terminal organ weights, or the plasma levels of liver enzymes. At concentrations where SRI-41315 was found to maximally efficacious, eRF1 protein levels were ~70% lower than the vehicle control. Please see the western-blot quantification data, Figure 6B. However, we observed toxicity with SRI-41315 beyond a concentration of 5µM, possibly due to excessively diminished levels of eRF1. We have also updated the results and discussion section to address this comment.

- It is hardly conceivable that a drug promoting eRF1 destabilization would specifically act on PTC without noticeable impact on NTC. This point must be clarified.

Response: Ribosomal profiling was performed to assess the effect of SRI-41315 on global translation termination compared to G418. We found that for cells treated with G418, we saw a global decrease in ribosomal density at stop codons with a corresponding increase in ribosomal density bound to 3' UTRs, indicating increased readthrough of stop codons globally. In contrast, for cells treated with SRI-41315, we found an increase of ribosomes bound to stop codons, indicating increased pausing at stop codons globally, but no increase in ribosomal density associated with 3' UTRs. The increased pausing at stop codons is consistent with a depletion of eRF1, which may be more pronounced at PTCs than NTCs due to more efficient stop codon recognition at normal stop codons due to surrounding mRNA context, proximity to the polyA tail, etc.

Reviewer 2.

This paper focuses on the mode of action the top hit identified in the accompanying paper (Du et al.), SRI-37240 and a more potent derivative, SRI-41315. With SRI-37240, the authors first report the dose-dependent increase of reporter readthrough, with a modest effect also on reporter mRNA abundance, and that it increases synergistically the conductance of a G542X mutant CFTR gene when combined with G418 or Amikacin, but not with Gentamycin. SRI-37240 was further tested on different CFTR nonsense mutations and in combination with G418, lumacaftor and ivacaftor. Applying SAR, they then derived SRI-41315 and showed that it has better physicochemical properties than SRI-37240 and that it is more potent in restoring NanoLuc activity and CFTR conductance in the respective reporter systems. Importantly, SRI-41315 also induced substantial readthrough in a human bronchial epithelial cell line, where SRI-37240 was essentially inactive on its own. When combined with G418, SRI-41315 restored 7-8% of WT CFTR activity and protein level in the HBEge G542X cells (Fig. 5). In Fig. 7, the authors further show that combined SRI-41315 and G418 also restored some CFTR activity in primary bronchial epithelial cells from a patient with nonsense mutations on both alleles of the CFTR gene, although this effect was more modest than in cells overexpressing CFTR reporter gene. Finally, the authors also reported the inhibition of other ion channels than CFTR by SRI-41315 and SRI-37240 (Supplemental information), which essentially invalidates the potential of these two molecules for becoming clinically applicable drugs.

Fig. 6 addresses the mode of action of the drug and appears a bit out of the context in this manuscript. Nevertheless, the finding that SRI-41315 causes a strong reduction of eRF1, presumably by its proteasome-mediated degradation (based on the sensitivity of this effect to MG132), is interesting. Depletion of eRF1 has previously shown to lead to increased readthrough, and since this is mechanistically different from the way aminoglycosides induce readthrough, the finding nicely explains the observed synergistic effects of combining SRI-41315 or SRI-37240 with G418.

Specific points to address:

- Refs 37-39 refer to abstracts from a congress in 2017, in which this screen is not described in sufficient detail. Instead, the screen should be described in detail in the accompanying paper by Du et al. and this manuscript here can then refer to Du et al.

Response: We have combined the Du et al manuscript in the resubmission, and accordingly, provided details on the screening methods, including data presented in Figure 1 and Supplementary Figure 1 and 2. We have updated the referencing accordingly.

- Lines 128-130: This statement needs a reference or the G418 data should be provided.

Response: This data has been removed from the current version to limit the number of figures in the combined manuscript.

- Lines 176-181: Comparison of the effect of SRI-37240 on WT (Fig. E1C, D) and G542X CFTR (Fig. 2B, C) indicates that SRI-37240 alone in both cases increases the conductance by about 3-fold. It is only when combined with G418, that there is a strong synergistic increase in the G542X but not in the WT. Do the authors have an explanation for this surprising data?

Response: This finding is consistent with the hypothesis that synergistic activity between G418 and SRI-37240 is dependent on the presence of a PTC, and unrelated to other aspects of CFTR expression or function present in the WT control.

- Figs. 2D, G, 3B and 5C: Please describe what Band C and Band B represent on these western blots.

Response: We have defined in the text so the importance and significance of visualizing CFTR band B and C can be better ascertained.

- Fig. 5C: Why is there a CFTR Band C in the lane from G542X untreated cells (lane 2), which is almost of the same intensity than when cells were treated with G418? If I understand correctly the genetic configuration of these cells, the only full-length CFTR that can be produced must originate from suppression of the nonsense codon at position 542, which is very unlikely to occur almost at the same rate under physiological conditions than when cells are treated with G418 (>1 or 2% of WT).

Response: We thank the reviewer for this comment. We have repeated this western blot about 7 times and by two different individuals at different laboratories (at UAB and CFFT laboratory). The band in question is reproducibly seen just above the CFTR band C position in 16HBE cells. This is seen not only with G542X but for other mutations as well. For example, we have tested 16HBEge R553X in parallel (please see the unpublished data below). We interpret this as a nonspecific band detected by the UNC 596 antibody. We also see this band with the exact same antibody in CFTR knockout rats, where no CFTR expression is present by mRNA, confirming this is a non-specific band (see Birket et al., Amer J Resp Crit Care Med 2020; and Sharma et al., Frontiers, 2020 in press).

- Line 315: Presumably, the authors wanted to say "...the possibility of decreased antibody avidity to eRF1 by SRI-41315".

Response: Correct. We have updated the text accordingly.

- The data shown in Figure 7C is not referred to in the text.

Response: We have updated the text to comment on this Figure panel.

Reviewer 3.

The authors demonstrate that a translational readthrough (RT) compound derived from an empiric screen acts synergistically with an established RT agent (G418). A second compound derived from the first shows somewhat higher effectiveness. However, as acknowledged by the authors, the new RT compounds are of limited clinical utility for CF due to off-target effects. The manuscript is an excellent example of the process for identifying, characterizing and reformulating RT compounds; however, the lack of clinical utility substantially decreases the impact of your findings. The manuscript could be considerably enhanced by investigating the mechanism of ENaC inhibition by the RT compounds and/or the feasibility of targeting eRF1 to antagonize nonsense-mediated decay.

General comment: Key aspects of the experimental design and rationale are missing in the text. For example, the reader has to search the legend and/or methods to determine which functional assay system is being used and when. Also, the switch between cDNA (not subject to NMD) and gene-based systems should be explained earlier in the results section.

Response: We are grateful for the overall favorable assessment of the work. In the revised version, we have attempted to be more clear when experiments transition between model systems, and the relevance of those systems to the biology of the research question.

Specific comments:

1) Figure 1: Several constructs are employed that are described in a prior publication. It would be helpful if a plasmid map was provided in this manuscript rather than requiring the reader to refer to another publication.

Response- We appreciate the reviewer for this comment. For the reasons mentioned above, we have combined the two manuscripts together so that the relevant reporters are described and presented.

2) Amelexanox is misspelled

Response: This data has been removed in the combined manuscript, and additional typographical errors corrected.

3) Why is reporter signal so much higher with NMD compared to RT construct?

Response: We expect the difference is related to the incomplete efficiency of NMD as opposed to translation termination, although site of integration may have also been a factor. Nevertheless, as noted in the responses above, we have removed the reporters that assess RT and NMD individually, and focused on the combined reporter data which drove the discovery and characterization of the agents in the report.

4) SRI-37240 alone and with G418 generate dramatic increase in luciferase expression. Why is the increase in expression of CFTR with these compounds so much lower?

Response: We appreciate the reviewer for this comment. However, the luciferase reporter optimized and chosen for its sensitive readthrough effect. While the unknown features of CFTR mRNA limits the readthrough effects in CFTR specific model systems.

5) Does Figure 3 use transient expression of CFTR variants as compared to stable expression of G542X in figure 2? Why this switch?

Response: We have clarified the confusion in the text. The cells used in former Figure 2 and 3 (revised Figure 2 and Supplementary Figure 4) all have stable expression of G542X.

6) Was the RT/RMD vector the transiently expressed in [16HBE14o-] cells in Figure 4.... are these the parental cells or the genetically modified cells used in Figure 5?

Response: We thank the reviewer for this comment. These cells are parental, genetically unmodified cells and we have modified the text accordingly.

7) Why was the R154X reporter used to test effect of manipulating eRF1 levels? Were effects different in those expressing CFTR with W1282X or G542X?

Response: These studies were tested in 16HBE G542X cells *via* the TECC assay (Figure 5A, B in the revised manuscript) along with NanoLuc reporters in the same cell type (revised Figure 4D,E). We also repeated the eRF1 depletion experiment with the W134X reporter instead of the W154X construct (Fig. 6D) and obtained similar results. We also edited the text to resolve any confusion.

8) Since proteasome inhibitors have broad effects, was any attempt made to selectively reduce eRF1 expression using siRNA or genetic knockout of eRF1?

Response: We appreciate the reviewer for this comment and agree that selective eRF1 reduction using siRNA will strengthen our mechanistic understanding. In the revised version we have performed the eRF1 siRNA-mediated knockdown and measured both NanoLuc and CFTR expression and function respectively. Similar to our findings using proteasome inhibitors, we confirm synergistic readthrough activity when eRF1 expression is genetically suppressed. Please see Figure 6 D-J.

Reviewers' Comments:

Reviewer #1:

Remarks to the Author:

This new version of the manuscript, by the Bedwell and Rowe teams, responds satisfactorily to the comments of the various reviewers. This new version, in which the two original papers have been merged, is of much better quality. The addition of new data as well as the enrichment of the discussion make this work very convincing and of very good quality.

As the authors describe it very well, these molecules are not necessarily of therapeutic interest by themselves, but by demonstrating that eRF1 is a therapeutic target of interest, this work opens up a significant therapeutic potential in the treatment of genetic diseases due to nonsense mutations. The work has been done rigorously and the explanations given are clear and precise.

I congratulate the authors for this impressive work.

I have only one minor remark.

Concerning the way in which SRI molecules induce the degradation of eRF1 by the proteasome. Given that structural data from eRF1 are available, have the authors looked at whether there is an easily predictable cavity or a binding motif within eRF1? The question behind this one is whether the two SRI molecules induce eRF1 degradation by a direct binding or in a more indirect way? This could be interesting to discuss this point in the discussion.

Olivier Namy

Reviewer #2:

Remarks to the Author:

The authors have followed the suggesting of the editor to combine the two initial manuscripts into one manuscript comprising a coherent story, and hence the revised combined manuscript has been almost completely rewritten.

It starts with describing the NanoLuc NMD/readthrough reporter and its use for a HTS of 771'345 compounds (Figs. 1, S1 and S2) and then focuses on two compounds (SRI-37240, covered by Novartis patent WO2014091446) and its derivative SRI-41315 (It is not clear to this reviewer if the derivative compound is also covered by the Novartis patent, which should be clarified and accordingly reported in the manuscript). They show that in FRT cells expressing a PTC-containing cDNA of human CFTR, SRI-37240 increased chloride conductance and full length CFTR abundance (Figs. 2, S3, S4). While applied on its own, SRI-37240 induced 1.4% of forskolin-stimulated conductance of the G542X CFTR mutant compared to wild-type CFTR, it showed a synergistic effect when combined with G418, reaching 10.5% of wild-type CFTR. Similar results were obtained with other PTCs in CFTR (R553X, R1162X, W1282X). Moving towards elucidating the mode of action of SRI-37240, the authors then present ribosome profiling data showing that opposite to G418, SRI-37240 did not induce general readthrough of stop codons but rather a caused prolonged ribosome stalling at stop codons (Fig. 3). Next, the authors present as the result of their chemical refining of the original compound a derivative (SRI-41315) with improved ADME properties and a higher readthrough-promoting efficacy (Fig. 4). As for SRI-37240, SRI-41315 gives the highest increase in conductance and full length CFTR when combined with G418 in the FRT cells (Fig. S5) and in 16HBEge cells, which contain a single copy of the CFTR gene with the G542X mutation, the combination of SRI-41315 with G418 restores 8% of WT CFTR protein (Fig. 5). Regarding the mode of action of SRI-41315, the authors show that it dramatically reduces the cellular abundance of eRF1, which could be suppressed by the proteasome inhibitor MG132, suggesting that SRI-41315 induces the proteasomal degradation of eRF1 (Fig. 6 and S7). Finally, using HBE cells donated from a cystic fibrosis patient heterozygous for the CFTR R553X/W1282X mutations, the authors show that combined treatment with SRI-41315 and G418 resulted in a small but significant increase of forskolin-stimulated current compared to G418 alone (Fig. 7), but they also report deleterious effects of SRI-41315 and SRI-37240 on ion conductance mediated by channels other than CFTR, which limits their potential for future clinical use to treat cystic fibrosis.

Specific points to address:

- 1.) I maintain my initial criticism that there is not enough details given about the HTS and its results. For example, the description of the library used for the screen and the origin of its compounds is missing. From the description in Methods (lines 528-529), it is not clear whether the described SRI-37240 compound was added to the library because of prior knowledge from Novartis about its readthrough activity (see patent WO2014091446 from 2014) or whether it happened to be present in this library and was independently of Novartis rediscovered as a readthrough promoter. While I agree with the authors response to Reviewer 1 that the information about SRI-37240 is not easily accessible in the Novartis patent, it is nevertheless crucial to give correct credit to the actual discoverers of the compound's activity.
- 2.) The ribosome profiling data shows that SRI-37240 induces no general readthrough of stop codons (Fig 3, metagene analysis), yet it quite efficiently promotes readthrough at CFTR PTCs (Fig 2). Can readthrough at PTC-containing endogenous transcripts be detected by in the ribo-seq data? Is there a link to NMD sensitivity, i.e. does it make a difference if the PTC-containing transcripts are NMD targets or not? In general, much more interesting information could be retrieved from the ribosome profiling data set than the rather superficial analysis presented in the manuscript.
- 3.) Fig 4E and p. 11, lines 249-255: Addition of G418 causes an about 2 to 3-fold increased NanoLuc activity than SRI-41315 alone, which I assume might be also the effect caused by G418 alone, in which case the effect is not synergistic, but additive. However, this cannot be judged by the reader, because the data for G418 alone in the 16HBE14o cells is missing.
- 4.) Fig. 6D and p. 15, lines 331-334: How do the authors explain the big difference between the almost complete protection of SRI-41315-induced eRF1 reduction by MG132 (Fig. 6C) and the resulting very modest change in NanoLuc readthrough?

Minor points:

- In many of the graphs, the different blue/turquoise color tones are difficult to distinguish given the small size of the printed figures. Please adjust the color scheme to make clearer distinctions. This applies to Figs. 1D, 2AB, 4CD, 5A, 6I and 7A.
- Line 425: delete "recently submitted", the patent was submitted 2014, 6.5 years ago.
- Author contributions: No contributions are listed for the co-authors Jamie Wangen and Rachel Green. I believe they provided the ribosome profiling data.

Reviewer #3:

Remarks to the Author:

This is a very significantly revised manuscript that addresses all the concerns I raised in my first review.

Reviewer #4:

Remarks to the Author:

The current version combining two prior submissions presents a logical flow of experimental work that should be accessible to the broad readership of Nature Communications. The work elucidating the effects of the readthrough compound on eRF1 are thorough and should be useful for investigators working on nonsense mediated decay and therapy of mutations that produce premature termination codons.

Specific comments:

- 1) Why did the authors not further investigate the mechanism by which the RT compounds inhibited sodium and CFTR currents as this is stated to be a major limitation of its clinical usefulness.

- 2) Given that hyperabsorption of sodium in the airway is believed to be a key component of CF pathophysiology, had the authors considered that the inhibitory effect of the RT compound might be advantageous in CF?
- 3) Figure 2. Sequence context is important for the action of readthrough compounds. SRI-37240 was identified based on a UGAA Sequence. Why was G542X selected that has a UGAG sequence?
- 4) Supplemental figure 4 shows differing effects of SRI 37240 on three different CFTR2 nonsense mutations. W1282X appears to be most responsive to the readthrough compound. The authors suggest that this is due to partially functional truncated CFTR, but could the UGAA sequence of the W1282X mutation be a better target for the RT compounds than the UGAG sequences of R553X and R1162X?
- 5) Figure 2. SRI-37240 combined with G418 generated 25% of wild type and band C CFTR but only 10% of wild type CFTR current. Is this due to the alternate amino acid inserted at codon 542?
- 6) Figure 5. The 16 HBEge cell line genome has at least two copies of CFTR. One copy is non-functional due to insertion of an SV40 sequence and inactivation of the synthesized RNA. This information should be corrected in the text of the results.
- 7) References 42 and 43 are the same.
- 8) To broaden interest in their work, it is suggested that the authors discuss the therapeutic potential of targeting eRF1 beyond CF (eg PMID 33009412 and 32059759, in addition to the hemophilia citation)

Response to reviewers for NCOMMS-20-12893A

Reviewer #1

- The work has been done rigorously and the explanations given are clear and precise. I congratulate the authors for this impressive work.

Response: We thank the reviewer for this positive comment.

Minor remark #1: Concerning the way in which SRI molecules induce the degradation of eRF1 by the proteasome. Given that structural data from eRF1 are available, have the authors looked at whether there is an easily predictable cavity or a binding motif within eRF1? The question behind this one is whether the two SRI molecules induce eRF1 degradation by a direct binding or in a more indirect way? This could be interesting to discuss this point in the discussion.

Response: There is not an obvious binding site on eRF1 for these molecules. This is not surprising, since our compounds may act in a manner analogous to the small molecule CC-885 that targets “neo-substates” such as Ikaros and GSPT1 to the E3 ubiquitin ligase cereblon (see doi:10.1038/nature18611). Neither of these neo-substates have appreciable affinity for the E3 ligase in the absence of this small molecule, which has been referred to as “molecular glue”. We have added a few sentences in the discussion to note this possibility.

Reviewer #2

- It starts with describing the NanoLuc NMD/readthrough reporter and its use for a HTS of 771'345 compounds (Figs. 1, S1 and S2) and then focuses on two compounds (SRI-37240, covered by Novartis patent WO2014091446) and its derivative SRI-41315 (It is not clear to this reviewer if the derivative compound is also covered by the Novartis patent, which should be clarified and accordingly reported in the manuscript).

Response: The rights to SRI-37240 and its derivative, SRI-41315 are both protected by the Novartis patent.

1.) I maintain my initial criticism that there is not enough details given about the HTS and its results. For example, the description of the library used for the screen and the origin of its compounds is missing. From the description in Methods (lines 528-529), it is not clear whether the described SRI-37240 compound was added to the library because of prior knowledge from Novartis about its readthrough activity (see patent WO2014091446 from 2014) or whether it happened to be present in this library and was independently of Novartis rediscovered as a readthrough promoter. While I agree with the authors response to Reviewer 1 that the information about SRI-37240 is not easily accessible in the Novartis patent, it is nevertheless crucial to give correct credit to the actual discoverers of the compound's activity.

Response: We apologize for misunderstanding the reviewer's previous request. We have now included the following information in the Methods section. “A collection of 771,345 unique, non-proprietary compounds custom assembled at Southern Research from various commercial vendors (ChemBridge, Enamine, ChemDIV, Life Sciences, Tripos) was screened in HTS. The compounds were selected to have

molecular properties matching eight criteria for lead-like molecules to serve as starting points for a drug discovery effort (Molecular Weight ≤ 500 ; Heteroatom count ≤ 10 ; Number Rotatable Bonds ≤ 8 ; Number Aromatic Rings ≤ 4 ; ALog P ≤ 6 ; Molecular Polar Surface Area ≤ 200 ; H-bond acceptors < 10 ; H-bond donors < 5). The collection is diverse, containing over 37,000 clusters (Tanimoto distance < 0.2) and over 275,000 non-overlapping Murcko scaffolds (determined by locating all ring systems of the molecule and all direct connections between them), including over 9,000 individual ring systems and approximately 16,000 contiguous ring systems with unique substitution pattern.

SRI 37240 was one of the compounds in the Enamine library (T6740530). Notably, we were not aware of the previous patent from Novartis regarding the readthrough properties of this compound at the time of the screen. Based on the structure of SRI-37240 compound, SRI-41315 was synthesized at Southern Research as part of the medicinal chemistry optimization (as stated in the manuscript). It was not part of the screening collection or guided by the patent contents.”

2.) *The ribosome profiling data shows that SRI-37240 induces no general readthrough of stop codons (Fig 3, metagene analysis), yet it quite efficiently promotes readthrough at CFTR PTCs (Fig 2). Can readthrough at PTC-containing endogenous transcripts be detected by in the ribo-seq data? Is there a link to NMD sensitivity, i.e. does it make a difference if the PTC-containing transcripts are NMD targets or not? In general, much more interesting information could be retrieved from the ribosome profiling data set than the rather superficial analysis presented in the manuscript.*

Response: Unfortunately, detection of stop codon readthrough at endogenous premature termination codons (PTCs) is not currently possible given the limitations of ribo-seq. Extensive research in this area have classified endogenous classes of nonsense-mediated decay (NMD) substrates (those stabilized by deletion of UPF factors in genome wide RNA-seq experiments) including mRNAs with PTCs derived from transcriptional errors, alternatively spliced transcripts that contain so called “poison-cassette” exons, and transcripts with long 3’UTRs. Due to the short fragment size of ribosome footprints (RPFs) and the complexity of alternative splicing in mammals, transcript-level assignment cannot be performed deterministically. This means that for the vast majority of RPFs that translate a transcript isoform containing a PTC, we cannot determine whether the upstream or downstream RPFs were actually translating the PTC-containing transcript (or a different isoform) -- downstream RPFs mostly represent the normal translation of alternate transcripts rather than the readthrough product of a PTC. Further, sequencing coverage of these PTC-containing transcripts is generally far too low given the degradation of these mRNAs by NMD.

Finally, widespread stop codon readthrough of endogenous PTCs should not be expected to shield these transcripts from NMD, as virtually all of these transcripts contain subsequent PTCs, because these sequences have not been selected as coding sequences. The probability of iterative readthrough events on these transcripts is vanishingly low and thus these substrates are not broadly stabilized (our analysis of our data bears this

out, see Response to Reviewers, **Figure 1**). By contrast, readthrough of typical disease-causing nonsense mutations, which would only require a single readthrough event to produce a full-length protein, effectively block these mRNAs from NMD.

3.) *Fig 4E and p. 11, lines 249-255: Addition of G418 causes an about 2 to 3-fold increased NanoLuc activity than SRI-41315 alone, which I assume might be also the effect caused by G418 alone, in which case the effect is not synergistic, but additive. However, this cannot be judged by the reader, because the data for G418 alone in the 16HBE14o cells is missing.*

Response: Figure 4E shows dose responses of SRI-37240 and SRI-41315 in the absence or presence of a constant amount of G418 (100 µg/ml). The data for G418 alone is represented by the first bar of the second and fourth groupings (0 µM of either SRI compound).

The log scale on the y-axis makes it harder to visualize the synergistic effect. To explain our logic, take SRI-37240 as an example. The maximum NanoLuc RLU resulting from SRI-37240 treatment alone was $\sim 2 \times 10^5$ RLU and for G418 treatment alone (first green bar) was $\sim 2 \times 10^6$ RLU. In contrast, the response when cells were treated with both compounds was $> 1 \times 10^7$ RLU, a value much larger than the 2.2×10^6 RLU that would be expected from an additive response. A similar analysis was done for cells treated with SRI-41315, leading us to conclude that the combined responses were synergistic.

4.) *Fig. 6D and p. 15, lines 331-334: How do the authors explain the big difference between the almost complete protection of SRI-41315-induced eRF1 reduction by MG132 (Fig. 6C) and the resulting very modest change in NanoLuc readthrough?*

Response: The western blots were done on extracts from cells treated with 10 µM MG132 to maximize the effect on eRF1 abundance. When we titrated the MG-132 concentration for the NanoLuc assays, we found that 10 µM MG132 caused a larger reduction in readthrough but also led to a modest reduction in WT NanoLuc activity, suggesting that concentration may induce a modest inhibition of translation. To avoid that complication, we reduced the MG132 concentration to 5 µM. At that dose, we did not see any reduction in WT NanoLuc activity (see lower part of Figure 6D). However, we still observed a smaller, but still significant, reduction in readthrough.

Minor points:

- *In many of the graphs, the different blue/turquoise color tones are difficult to distinguish given the small size of the printed figures. Please adjust the color scheme to make clearer distinctions. This applies to Figs. 1D, 2AB, 4CD, 5A, 6I and 7A.*

Response: Thanks for pointing this out. We re-plotted the graphs with different colors and increased the size of the symbols to make it easier to distinguish the different lines.

- *Line 425: delete “recently submitted”, the patent was submitted 2014, 6.5 years ago.*

Response: The wording regarding the patent was revised as requested.

- *Author contributions: No contributions are listed for the co-authors Jamie Wangen and Rachel Green. I believe they provided the ribosome profiling data.*

Response: Thank you, this oversight was corrected.

Reviewer #3

-This is a very significantly revised manuscript that addresses all the concerns I raised in my first review.

Response: We thank the reviewer for their review.

Reviewer #4

1) Why did the authors not further investigate the mechanism by which the RT compounds inhibited sodium and CFTR currents as this is stated to be a major limitation of its clinical usefulness.

We feel that the mechanism by which other ion channels like ENaC are inhibited by SRI-41315 is beyond the scope of the current study.

2) Given that hyperabsorption of sodium in the airway is believed to be a key component of CF pathophysiology, had the authors considered that the inhibitory effect of the RT compound might be advantageous in CF?

Response: This is a reasonable thought and one that was considered. Ultimately, this proved problematic, as translational readthrough is envisioned as a systemic therapy, whereas ENaC inhibition for the treatment of cystic fibrosis is necessarily an inhaled therapy, to provide ENaC inhibition to the respiratory airways, but avoiding ENaC inhibition in the nephron, where it could potentially cause dangerous hyperkalemia. Further examination of the ENaC inhibitory action of these compounds will be needed to determine whether potential complications arise, and this is outside the scope of this study.

3) Figure 2. Sequence context is important for the action of readthrough compounds. SRI-37240 was identified based on a UGAA Sequence. Why was G542X selected that has a UGAG sequence?

Response: We chose to examine the responsiveness of the CFTR-G542X mutation (UGAG tetranucleotide) and surrounding codons because it is the most common CFTR PTC allele. It should be noted that the 4th position of the tetranucleotide can influence the magnitude of readthrough, but this generally shows a greater readthrough bias toward pyrimidines (C and U) vs purines (A and G). However, readthrough has been routinely observed with all 4 tetranucleotide termination signals. Since UGAA vs UGAG differ only by the purine in position 4, we were confident that readthrough would be seen for both. However, this subtle distinction could explain small differences in the magnitude of readthrough.

4) Supplemental figure 4 shows differing effects of SRI 37240 on three different CFTR2 nonsense mutations. W1282X appears to be most responsive to the readthrough compound. The authors suggest that this is due to partially functional truncated CFTR, but could the UGAA sequence of the W1282X mutation be a better target for the RT compounds than the UGAG sequences of R553X and R1162X?

Response: Yes, that is possible. However, as mentioned in the discussion, it has been shown in previous studies that CFTR correctors can partially rescue the W1282X cDNA in the absence of readthrough.

5) Figure 2. SRI-37240 combined with G418 generated 25% of wild type and band C CFTR but only 10% of wild type CFTR current. Is this due to the alternate amino acid inserted at codon 542?

Response: Yes, that is a possible cause for this difference. However, these western blots are difficult to quantitate (see the size of the error bars in Figure 2E for the western from the combined treatment). Thus, we were careful to not over-interpret that difference between activity and full-length protein.

6) Figure 5. The 16 HBEge cell line genome has at least two copies of CFTR. One copy is non-functional due to insertion of an SV40 sequence and inactivation of the synthesized RNA. This information should be corrected in the text of the results.

Response: We thank the reviewer for this comment. This point has now been clarified in the methods section.

7) References 42 and 43 are the same.

Response: The references have been corrected and reformatted.

8) To broaden interest in their work, it is suggested that the authors discuss the therapeutic potential of targeting eRF1 beyond CF (eg PMID 33009412 and 32059759, in addition to the hemophilia citation)

Response: Thanks for this comment. We have now mentioned the potential of readthrough agents targeting eRF1 for treatment of other genetic diseases caused by nonsense mutations besides CF and hemophilia.

Reviewers' Comments:

Reviewer #2:

Remarks to the Author:

In their response-to-authors letter, the authors have satisfactorily answered my questions and resolved my concerns revolving about the screen, correct citation of the patented compounds and the ribosome profiling experiment. I have no more queries and in my opinion, the manuscript in its current form is now ready for publication.

Reviewer #4:

Remarks to the Author:

Thank you for your thoughtful responses to my queries